# GluN2A and GluN2B NMDA receptors use distinct allosteric routes

Meilin Tian[1,3], David Stroebel [1,3], Laura Piot[1], Mélissa David[1], Shixin Ye[2,4✉] & Pierre Paoletti [1,4✉]

Allostery represents a fundamental mechanism of biological regulation that involves long-range communication between distant protein sites. It also provides a powerful framework for novel therapeutics. NMDA receptors (NMDARs), glutamate-gated ionotropic receptors that play central roles in synapse maturation and plasticity, are prototypical allosteric machines harboring large extracellular N-terminal domains (NTDs) that provide allosteric control of key receptor properties with impact on cognition and behavior. It is commonly thought that GluN2A and GluN2B receptors, the two predominant NMDAR subtypes in the adult brain, share similar allosteric transitions. Here, combining functional and structural interrogation, we reveal that GluN2A and GluN2B receptors utilize different long-distance allosteric mechanisms involving distinct subunit-subunit interfaces and molecular rearrangements. NMDARs have thus evolved multiple levels of subunit-specific allosteric control over their transmembrane ion channel pore. Our results uncover an unsuspected diversity in NMDAR molecular mechanisms with important implications for receptor physiology and precision drug development.

[1] Institut de Biologie de l'Ecole Normale Supérieure (IBENS), Ecole Normale Supérieure, Université PSL, CNRS, INSERM, Paris, France. [2] Unité INSERM U1195, Hôpital de Bicêtre, Université Paris-Saclay, Paris, Le Kremlin-Bicêtre, France. [3] The authors contributed equally: Meilin Tian, David Stroebel. [4] These authors jointly supervised this work: Shixin Ye, Pierre Paoletti. ✉email: shixin.ye-lehmann@inserm.fr; pierre.paoletti@ens.psl.eu

Glutamate is the principal excitatory neurotransmitter in the central nervous system of vertebrates. Once released in the synaptic cleft, glutamate mediates fast neurotransmission by acting on ionotropic glutamate receptors (iGluRs) which are subdivided into three main families[1–3]: AMPA, kainate and NMDA receptors. While AMPA and kainate receptors are primarily involved into glutamate-induced depolarization of the postsynaptic membrane, NMDARs act as key drivers of synaptic plasticity which is widely thought to be a cellular substrate for learning, memory, and cognition in general[1,4].

NMDARs are heterotratemeric assemblies typically composed of two GluN1 subunits and two GluN2 subunits, of which there are four subtypes (GluN2A-D)[5–9]. Each subunit displays a common membrane topology with a cytoplasmic C-terminal domain (CTD), a pore-forming transmembrane domain (TMD) and an extracellular region composed on two large bilobate (or clamshell-like) domains: the N-terminal domain (NTD) and the agonist or ligand-binding domain (LBD) binding glutamate in GluN2 subunits and the co-agonist glycine (or D-serine) in GluN1 subunits[6]. Crystal and cryo-EM structures of near full-length (CTD deleted; referred to "full-length" thereafter) GluN1/GluN2 receptors[10–17] reveal that NMDARs form massive molecular complexes (>400 kDa) adopting a layered architecture with at the "top" the NTDs, at the "bottom" the TMD and CTD, and lying in between the LBDs. Moreover, the receptor assembles as dimer-of-dimers, with an alternate GluN1-2-1-2 subunit arrangement around the central pore, and with the LBDs and NTDs locally forming pairs of 2-fold related GluN1/GluN2 heterodimers[2,18].

There is currently vivid interest in determining whether individual NMDAR subtypes carry out specific functions in the normal and diseased brain. The adult brain relies predominantly upon GluN2A and GluN2B subunits which assemble as diheteromeric GluN1/GluN2A and GluN1/GluN2B receptors, or as triheteromeric GluN1/GluN2A/GluN2B receptors[6,19,20]. At synaptic sites, NMDARs typically contain GluN2A subunits, display faster kinetics and mediate long-term synaptic plasticity. Extra-synaptic NMDARs typically are enriched in GluN2B subunits, display slower kinetics and are thought to constitute a major signaling pathway that triggers neuronal death (excitotoxicity), as occurring in Alzheimer's or Parkinson's disease[21–24]. If excessive NMDAR activation is harmful to neurons, so is too little. In particular, NMDAR hypofunction plays a critical role in the pathophysiology of schizophrenia[6,23,25,26]. Human NMDAR genes are highly intolerant to variations[27,28], and mutations in Grin2a and Grin2b, the genes encoding the GluN2A and GluN2B subunits, respectively, are a major cause of neurodevelopmental disorders associated with childhood epilepsy and cognitive impairments[28,29]. Understandably, major efforts are currently undertaken to correct NMDAR dysfunction using drug compounds or biologics that can either boost or dampen NMDAR signaling with high subunit-selectivity[30–34].

In agreement with the critical importance of fine-tuning their activity, NMDARs are endowed with exquisite allosteric capacity that has no equivalent in other iGluRs[35]. The NTDs of GluN2A- and GluN2B-containing receptors form major allosteric hubs that control critical receptor properties including channel maximal open probability (Po) and deactivation kinetics[36,37]. These domains also confer a rich pharmacology to NMDARs, through binding of a variety of exogenous and endogenous small molecule ligands[31]. Hence, both GluN2A and GluN2B NTDs form sensors to endogenous zinc, which is enriched and released at many excitatory synapses where it acts as a potent allosteric inhibitor of synaptic NMDARs. The GluN2 NTD-zinc interaction has profound consequences on CNS function with involvement in synaptic plasticity, pain processing and cognition[38–43]. Reciprocally, LBD conformational changes affect NTDs motions and binding of modulatory ligands. Therefore, the extracellular region of NMDARs acts an integrated allosteric unit, whereby the NTD and LBD layers reciprocally influence one another both structurally and functionally[31,44]. The extensive NTD-LBD interactions observed in NMDARs, unlike AMPA and kainate receptors where interactions are minimal[45,46], provide the physical substrate for this allosteric coupling[10,11,13–16,35,47].

GluN2A and GluN2B receptors share similar mechanisms at the level of individual NTDs. In particular, zinc binding at both GluN2A and GluN2B NTDs triggers comparable clamshell closure motions[15,16,36,48–50]. More generally, both receptor subtypes display comparable overall shape and subunit arrangement, with several important domain–domain interfaces highly conserved[10–17] (Supplementary Fig. 1). Because of these shared features, it is generally assumed that GluN2A and GluN2B receptor subtypes undergo similar long-distance allosteric transduction mechanisms. Here, combining cellular electrophysiology, engineering of unnatural amino acids and in silico analysis, we show that the two main NMDAR subtypes in the adult brain adopt different NTD-driven transduction pathways to regulate their ion channel activity. The conformational spread in GluN2A and GluN2B receptors involves different inter-subunit and inter-domain interfaces and conformational rearrangements. These results provide new insights on NMDAR subtype-specific mechanisms and dynamics. They also widen the pharmacological landscape of this important class of neuronal receptors opening opportunities for precise subunit-targeted therapeutic interventions.

## Results

**Trapping GluN2B receptors in a high Po state**. To understand how distinct NMDAR conformations relate to functional states, we designed a set of experiments aimed at trapping specific domains of the receptor in a particular conformation while recording their ion channel activity. For that purpose, we used a photochemical approach coupling cellular electrophysiology and genetic encoding of light-sensitive unnatural amino acids (UAAs). More specifically, we introduced the photo-cross-linker UAA p-azido-phenylalanine (AzF) at given positions in the targeted domains and recorded the receptor response while photostimulating. This approach provides a powerful mean to correlate in real-time functional changes with structural rearrangements caused by light-induced crosslinking[51–54]. Using the full-length GluN1/GluN2B receptor structure[10,11] as a guide, we selected eight positions in the lower lobes of GluN1 and GluN2B NTDs, a region that has been proposed to undergo large conformational rearrangements during allosteric signaling[13,15,16,47,50,55,56]. Positions lined the solvent-exposed faces of GluN1 and GluN2B NTD α5 helices, two α-helices facing each other but far apart in the structure of the inhibited receptor (Cα-Cα distance >17 Å between GluN1-K178 and GluN2B-N184, Refs. [10,13] and Fig. 1a). Following expression of individual AzF mutant GluN1/GluN2B receptors into Xenopus oocytes, we applied UV illumination directly onto the cells and checked for potential changes in receptor activity. Among the eight mutants, GluN1-K178AzF exhibited a remarkable light sensitivity with a massive potentiation of current amplitude after UV treatment during agonist exposure ($4.39 \pm 1.40$ fold potentiation [$n = 37$]; Fig. 1b). In contrast, no significant effect was observed when UV light was shined on control wild-type GluN1/GluN2B receptors ($1.04 \pm 0.09$ [$n = 16$]; Fig. 1b). Successive applications of agonists alone following UV application revealed that the photo-potentiation remained stable for prolonged periods of time (minutes) with no

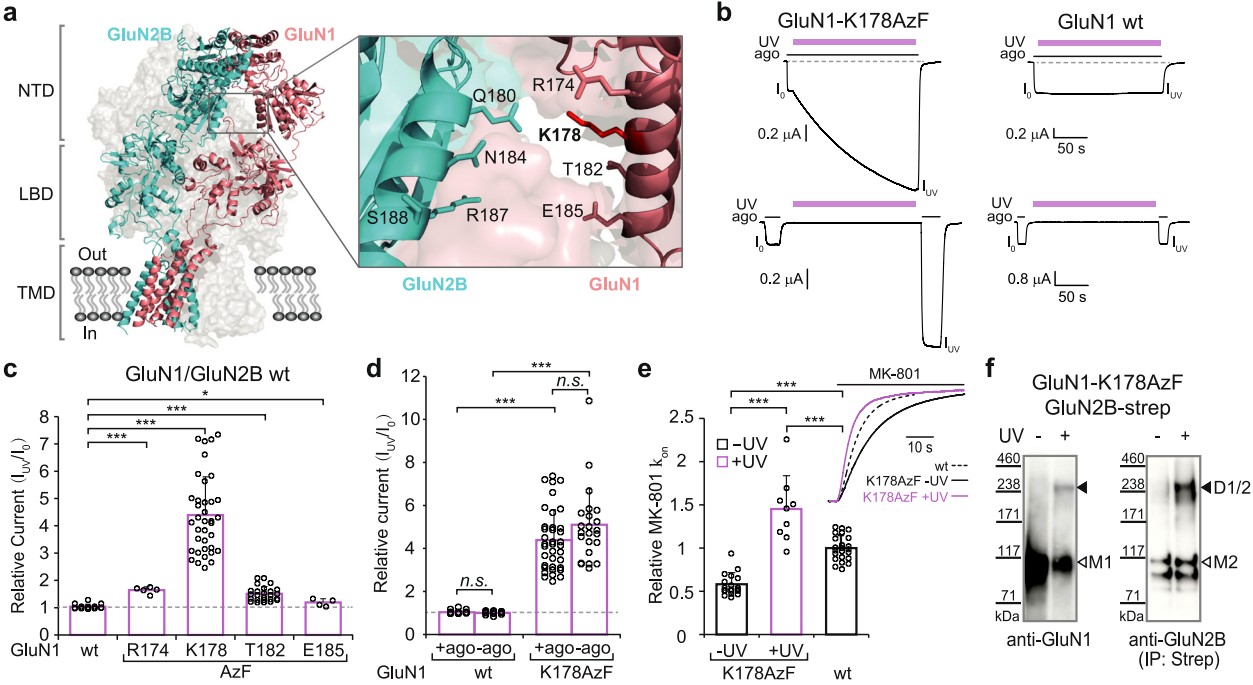

**Fig. 1 NTD heterodimer photocrosslinking locks GluN2B receptors in a high Po mode. a** Structure of the full-length GluN1/GluN2B receptor in the inhibited state (side view; PDB 4PE5, Ref. [10] and see "Methods"). The receptor, composed of two GluN1 subunits and two GluN2B subunits, assembles as a dimer-of-dimers and display a layered arrangement (NTD, LBD, TMD). The two front subunits are shown in cartoon representation while the two subunits in the back are displayed in space-filled. NTD N-terminal domain, LBD ligand-binding domain also named agonist-binding domain, TMD transmembrane domain. Inset: enlargement of the GluN1-GluN2B NTD heterodimer lower lobe region. Residues in GluN1 and GluN2B subunits subjected to amber mutation (allowing incorporation of photo-cross-linker amino acids) are represented as sticks. **b** Representative current traces from oocytes expressing GluN1-K178AzF/GluN2B mutant (left) and wild-type (wt, right) receptors during UV illumination (365 nm) in the presence (top) or absence (bottom) of agonists. ago, for agonists (glutamate and glycine, 100 μM each). **c** Change in current amplitude upon UV illumination ($I_{uv}/I_o$) of wild-type (wt) GluN1/GluN2B receptors and various GluN1-AzF/GluN2B mutant receptors. Values are: $1.04 \pm 0.09$ ($n = 16$) for wt, $1.65 \pm 0.12$ ($n = 5$) for GluN1-R174AzF, $4.39 \pm 1.40$ ($n = 37$) for GluN1-K178AzF, $1.51 \pm 0.26$ ($n = 25$) for GluN1-T182AzF and $1.19 \pm 0.13$ ($n = 4$) for GluN1-E185AzF. Data represent mean ± SD. $n$ = number of biologically independent cells. *$P = 0.011$, ***$P < 0.001$ (one-way ANOVA). **d** Change in current amplitude upon UV illumination ($I_{UV}/I_o$) of wt and GluN1-K178AzF/GluN2B mutant receptors during (+ago) or between (−ago) agonist application. Values are, from left to right: $1.04 \pm 0.09$ ($n = 16$) for wt +ago; $1.00 \pm 0.07$ ($n = 14$) for wt −ago; $4.39 \pm 1.40$ ($n = 37$) for mutant +ago; $5.05 \pm 1.73$ ($n = 23$) for mutant −ago. Data represent mean ± SD. $n$ = number of biologically independent cells. ***$P < 0.001$, n.s. non-significant ($P = 0.25$ for wt ± ago, and 0.11 for GluN1-K178AzF ± ago) (one-way ANOVA). **e** Assessment of receptor channel activity using MK-801 inhibition kinetics. MK-801 $k_{on}$ values were normalized to the mean value obtained with wild-type (wt) GluN1/GluN2B receptors. Relative values are, from left to right: $0.57 \pm 0.13$, ($n = 19$) without UV and $1.45 \pm 0.38$ ($n = 9$) with UV for GluN1-K178AzF/GluN2B; $1.00 \pm 0.15$ ($n = 23$) for wt (no UV). Inset: representative scaled current traces from oocytes expressing wild-type (dashed) or GluN1-K178AzF/GluN2B receptors before (black) and after (violet) UV illumination in response to 50 nM MK-801. Data represent mean ± SD. $n$ = number of biologically independent cells. ***$P < 0.001$ (one-way ANOVA). **f** Immunoblots from HEK cells expressing GluN1-K178AzF/GluN2B mutant receptors and exposed to UV (+) or not (−). Samples were analyzed using anti-GluN1 and anti-Strep antibodies. GluN1 monomer (M1) is expected to run at ~110 kDa, GluN2B-Strep monomer (M2) at ~130 kDa, and GluN1/GluN2B heterodimer (D1/2) at ~240 kDa. "IP: Strep" indicates the treatment with an immuno-purification procedure using the Strep tag.

sign of reversibility (Supplementary Fig. 2a), compatible with the formation of a covalent bond. Besides position GluN1-K178, six additional positions in either GluN1 or GluN2B subunits yielded UV-induced photo-potentiation, although of smaller amplitude (Fig. 1c and Supplementary Fig. 2b). For all mutants, we verified that no or tiny current responses were observed in AzF-deprived incubation medium (while large currents were readily observed in the presence of AzF; Supplementary Fig. 3a), confirming the successful and efficient incorporation of AzF by the orthogonal aaRS/suppressor tRNA pair.

Illuminating GluN1-K178AzF/GluN2B receptors in the resting state (i.e., in the absence of agonist), produced strong photo-potentiation similar in amplitude to that observed on the active state ($5.05 \pm 1.73$ potentiation [$n = 23$]; Fig. 1b, d), revealing that the photoreactive effect occurs independently of the functional state of the receptor. We further characterized the photo-responsiveness of GluN1-K178AzF/GluN2B receptors by

assessing the channel maximal Po before and after UV treatment. Toward this end, we measured inhibition kinetics by the NMDAR open-channel blocker MK-801, a proxy classically used to estimate channel Po[47]. We found that before UV treatment, GluN1-K178AzF/GluN2B receptors displayed significantly slower MK-801 inhibition kinetics compared to wild-type receptors (relative MK-801 $k_{on} = 0.57 \pm 0.13$ [$n = 19$]; Fig. 1e), indicative of a decreased Po. Following UV treatment, however, mutant receptors showed a marked increase in channel Po, as evidenced by the large increase in MK-801 on-rate inhibition kinetics, reaching a value surpassing that of wild-type receptors ($1.45 \pm 0.38$, [$n = 9$]; Fig. 1e). Similar trends were also observed for GluN1-R174AzF and GluN1-T182AzF mutants, both potentiated by UV, but not for the GluN1-E185AzF mutant showing minimal sensitivity to UV illumination (Fig. 1c and Supplementary Fig. 3b). Whole-cell patch-clamp experiments performed on HEK cells expressing GluN1-K178AzF/GluN2B receptors fully

confirmed the large photo-potentiation effects observed in *Xenopus* oocytes (Supplementary Fig. 4a–b). Moreover, single-channel recordings combined to noise analysis revealed that the photo-potentiation is fully accounted by an increase in channel Po with no change in channel unitary conductance, and that cross-linked receptors are "locked" in a high Po mode (Supplementary Fig. 4c–e). Surprisingly, no UV-dependent photo-modulation was observed when incorporating the photo-cross-linker UAA p-benzoyl-L-phenylalanine (BzF) at any of the four GluN1 sites despite evidence indicating successful BzF incorporation (Supplementary Fig. 5a). Hence, at these GluN1 NTD lower lobe sites, the UV-dependent potentiation is specific for the AzF photochemistry.

Because the long side chain of GluN1-K178 points toward the neighboring GluN2 NTD lower lobe (Fig. 1a), we hypothesized that UV treatment of GluN1-K178AzF/GluN2B receptors results in the crosslinking of GluN1 and GluN2B subunits. Western blot experiments confirmed this prediction, showing a clear high molecular weight band recognized by both GluN1 and GluN2B antibodies from UV-treated receptors but not untreated ones (Fig. 1f). We obtained additional evidence that AzF photo-crosslinking was essential to the photomodulation rather than alternative non-crosslinking azido associated photochemical pathways (including formation of amines; Supplementary Fig. 5b and see Ref. [57]), by performing thiol-modifying experiments on GluN1-K178C/GluN2B receptors. Grafting the positively charged amine compound MTSEA or MTSET on GluN1-K178C produced no effect on receptor activity, in striking contrast to the large potentiation observed following UV treatment of the GluN1-K178AzF mutant (Supplementary Fig. 5c). Taken together, these results show that AzF is an efficient bio-orthogonal tool for remote manipulation of NMDAR inter-subunit interfaces. They also show that the conformational state of the NTD heterodimer has major influence on GluN2B receptor activity, in line with previous results [13,55]. Trapping the NTD dimer in a compact conformation in which the NTD lower lobes are close to one another locks the receptor in a high Po mode, facilitating channel opening.

**Photopotentiation requires inter-subunit mobility.** The GluN1/GluN2 NTD heterodimer is a highly dynamic structural element of GluN2A and GluN2B NMDARs that can undergo both intra-domain and inter-domain conformational changes [13,15,16,47,50,56,58,59]. To determine which of these intra- or inter-subunit motions are important for the photo-reactivity of GluN1-AzF/GluN2B receptors, we repeated the UV photo-crosslinking experiments on receptors with structurally constrained NTDs. We thus coupled the GluN1-K178 AzF mutation with a series of disulfide bond mutations previously shown to lock either the GluN1 NTD in a closed cleft conformation (GluN1-S126C-H171C also named N1-CC), or the GluN2B NTD in a closed cleft conformation (GluN2B-A135C-P177C also named 2B-CC), or to restrict inter-domain reorientation (GluN1-L320C/GluN2B-D210C also named N1-C/2B-C) (Fig. 2a and Supplementary Fig. 6; Refs. [56,60]). As illustrated in Fig. 2b–c, both receptors with individual NTDs "locked closed" still showed UV current potentiation, although of lesser amplitude than the parent GluN1-K178AzF/GluN2B receptors ($1.63 \pm 0.29$ [$n = 9$] for N1-CC; $2.52 \pm 0.21$ [$n = 3$] for 2B-CC). In contrast, UV sensitivity was almost fully abolished at N1-C/2B-C receptors which behaved like wild-type GluN1/GluN2B receptors ($1.12 \pm 0.10$ [$n = 8$] for N1-C/2B-C vs $1.04 \pm 0.09$ [$n = 16$] for WT receptors, $P > 0.05$; Fig. 2b–c). Altogether, these results establish that conformational mobility of the NTDs is required to permit photo-crosslinking and switch the receptors in a high Po mode.

Specifically, they indicate that positive modulation through the NTDs involves substantial rearrangement of the NTD hetero-dimer packing with openings of the individual NTD clamshells and displacement of GluN1 and GluN2B NTD lower lobes in closer proximity.

**NTD trapping differentially impacts GluN2A and GluN2B receptors.** The GluN2 NTDs behave in a semi-autonomous manner, retaining their functional properties when transplanted from one GluN2 subunit to another [36,37,55,61–63], possibly arguing for conserved allosteric mechanisms between NMDAR subtypes. With this premise in mind, we decided to systematically compare how GluN1/GluN2B and GluN1/GluN2A receptors respond to manipulations aimed at restraining their conformational mobility. We first replicated the above-described photo-crosslinking experiments on GluN2A receptors using the same photo-reactive GluN1-AzF NTD mutations. Similar to what we observed on GluN2B receptors, UV illumination resulted in potentiation of current responses for all four GluN2A mutants, being highest for GluN1-K178AzF/GluN2A receptors (Fig. 3a and Supplementary Fig. 7a). For these later, the light-induced effect was accounted by an increased channel Po and showed similar time course than at GluN2B receptors (Supplementary Fig. 7b, c). It was also accompanied by the formation of an inter-subunit covalent photo-bridge between GluN1 and GluN2A subunits (Fig. 3a inset). Quantitatively, however, the extent of photo-modulation differed greatly from that observed on GluN2B receptors, being of much smaller amplitude on GluN2A receptors ($1.45 \pm 0.23$ [$n = 32$] fold potentiation for GluN1-K178AzF/GluN2A receptors vs >4-fold potentiation for GluN1-K178AzF/GluN2B receptors, Fig. 3a, b and Supplementary Fig. 4a, b). This difference is not unexpected given that GluN2A receptors have a much higher basal channel maximal Po than GluN2B receptors (0.5 vs 0.15; Refs. [36,64]), thereby reducing the range for potential increase in channel activity. Accordingly, we repeated the UV illumination experiments at acidic pH (pH 6.5), conditions under which all NMDAR subtypes are under strong allosteric inhibition by ambient extracellular protons [65–67]. Comparing the photo-responsiveness of GluN1-K178AzF/GluN2A and GluN1-K178AzF/GluN2B receptors revealed a strikingly different behavior between the two receptor subtypes. For GluN2B receptors, photo-potentiation at pH 6.5 was massive doubling in amplitude compared to that observed at pH 7.3 ($8.59 \pm 3.71$ [$n = 41$], Fig. 3b). Western blots confirmed the UV-induced GluN1-GluN2B inter-subunit attachment at low pH (Supplementary Fig. 7d). This result is consistent with proton inhibition and photo-crosslinking opposing one another, protons favoring the GluN1/GluN2B NTD heterodimer to adopt a loose (or relaxed) conformation which shifts the receptors in a low activity mode, while photo-crosslinking reverts the NTD heterodimer in a more compact conformation which traps the receptors in a high activity mode. In contrast, for GluN2A receptors, photo-potentiation at pH 6.5 remained remarkably small ($1.23 \pm 0.37$ [$n = 58$]; Fig. 3a), with amplitudes even lower than those observed at pH 7.3 (Fig. 3b) and despite the formation of a GluN1-GluN2A inter-subunit covalent bond (Fig. 3a inset). These results unveil that GluN2A and GluN2B receptors differ in their pH-sensitive allosteric machinery. In GluN2A receptors, allosteric inhibition through protons appears largely disconnected from the compactness of the NTD heterodimer, while in GluN2B receptors both are strongly coupled.

We obtained further evidence for dissimilar allosteric mechanisms between GluN2A and GluN2B receptors by comparing the impact of NTD photo-crosslinking on the receptor pharmacology. For that purpose, we determined the sensitivity of GluN2A

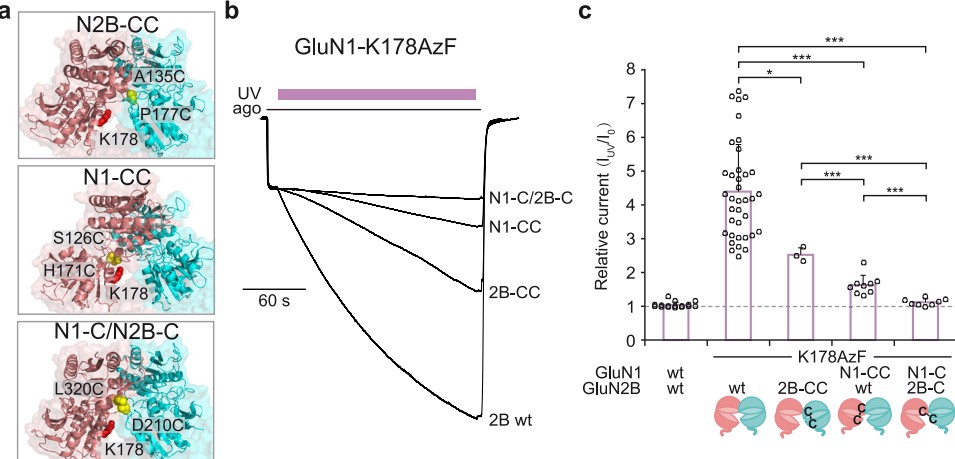

**Fig. 2 Photopotentiation necessitates inter-subunit mobility of the GluN1/GluN2B NTD heterodimer. a** Location of each double cysteine mutations (CC) in the GluN1/GluN2B NTD heterodimer (PDB 4PE5, Ref. [10]). Mutant receptors are GluN1-K178AzF/GluN2B-A135C-P177C (2B-CC), GluN1-S126C-H171C-K178AzF/GluN2B wt (N1-CC) and GluN1-K178AzF-L320C/GluN2B-D210C (N1-C/2B-C). Introduced cysteines are highlighted as yellow spheres and the GluN1-K178 position as red spheres. **b** Responsiveness of disulfide-bond linked mutant receptors to UV illumination (365 nm). Representative current traces measured from oocytes expressing wild-type (wt) or various cysteine mutant receptors (as described in a) during UV illumination. Note that the UV-induced potentiation is almost completely lost on receptors with inter-subunit disulfide bond cross-linked NTDs (N1C/2B-C). ago, for agonists (glutamate and glycine, 100 μM each). **c** Change in current amplitude upon UV illumination ($I_{uv}/I_o$) of wild-type (wt) GluN1/GluN2B receptors and cysteine mutant receptors (as described in a). Values are: $1.04 \pm 0.09$ ($n = 16$) for wt, $4.39 \pm 1.40$ ($n = 37$) for GluN1-K178AzF/GluN2B, $2.52 \pm 0.21$ ($n = 3$) for 2B-CC, $1.63 \pm 0.29$ ($n = 9$) for N1-CC and $1.12 \pm 0.10$ ($n = 8$) for N1-C/2B-C. Bottom cartoons illustrate the introduced disulfide bond pair and their impact on NTD clamshell closure. Data represent mean ± SD. $n$ = number of biologically independent cells. $*P = 0.030$, $***P < 0.001$ (one-way ANOVA).

and GluN2B receptors to several allosteric modulators before and after UV treatment, once the receptors are trapped in a high Po mode. Exploiting the rich pharmacology endowed by the NTDs[31], we tested the sensitivity to the endogenous modulators zinc and protons, as well as to spermine and ifenprodil, two GluN2B-selective allosteric modulators[1,5,31]. Full zinc dose-response curves of GluN1-K178AzF/GluN2A receptors revealed that the zinc potency (i.e., $IC_{50}$) was virtually unchanged following UV treatment ($IC_{50} = 7.43 \pm 1.10$ nM [$n = 4$] pre-UV vs $7.69 \pm 0.17$ nM [$n = 3$] post-UV, $P > 0.05$, respectively; Fig. 3c left panel), while the maximal level of zinc inhibition was modestly decreased from 71% to 52%. Assessment of proton sensitivity revealed even less effect of UV treatment on GluN2A receptors, pH dose-response curves being almost indistinguishable between the two conditions (pH $IC_{50}$ of $7.0 \pm 0.02$ [$n = 9$] pre-UV vs $7.01 \pm 0.01$ [$n = 10$], post-UV, $P > 0.05$; Fig. 3d left panel). Contrasting with this situation, strong effects were seen at GluN2B receptors. UV crosslinking of GluN1-K178AzF/GluN2B receptors induced a marked decrease in proton sensitivity, manifested by >0.5 pH-unit rightward shift in the pH dose-response curve (pH $IC_{50}$s of $7.60 \pm 0.01$ [$n = 7$–9] pre-UV vs $6.90 \pm 0.02$ [$n = 5$–6] post-UV, $P < 0.001$; Fig. 3d right panel). Similarly, UV treatment strongly decreased zinc sensitivity of GluN2B receptors, increasing zinc $IC_{50}$ by ~15-fold ($0.54 \pm 0.01$ μM [$n = 4$] pre-UV vs $7.96 \pm 0.16$ μM [$n = 3$] post-UV, $P < 0.05$; Fig. 3c right panel). This displacement in zinc sensitivity was in fact approaching that obtained when deleting the whole GluN2B NTD where zinc binds (Fig. 3c dotted line). Additional tests showed that UV illumination of GluN1-K178AzF receptors led to a pronounced decrease in the sensitivity to the allosteric inhibitor ifenprodil (Fig. 3e), as well as to a massive reduction in spermine potentiation (Fig. 3f). Overall, these results provide evidence that GluN2A and GluN2B receptors are differentially affected when imposing shared structural constraints on their individual NTD dimers. Locking GluN1/GluN2 NTD dimers in a compact conformation minimally alters allosteric modulation of GluN2A receptors but disrupts allosteric modulation of GluN2B receptors.

**NTD inter-dimer mobility is essential for GluN2A, but not GluN2B, receptor allostery.** Structural studies on full-length NMDARs indicate large conformational mobility within individual NTD heterodimers but also between the two constitutive NTD pairs. The two NTD heterodimers can be in close apposition making contacts through their GluN2 NTD lower lobes along the two-fold Z axis of symmetry of the tetrameric arrangement. Alternatively, the NTD region can adopt more "extended" conformations, in which the two NTD pairs have moved apart ablating this NTD inter-dimer tetramerization interface[11–13,15,16]. To evaluate whether NTD inter-dimer mobility is critical for NMDAR allosteric regulation, we used a disulfide bond strategy aimed at preventing separation between the two NTD dimers (Supplementary Fig. 6). Cysteines introduced at the homologous positions V217C in GluN2A NTD and N218C in GluN2B NTD induced the formation of a redox-sensitive disulfide link between the two GluN2 NTDs in both GluN2A and GluN2B receptors (Fig. 4a, b), in agreement with previous reports[10,11,47]. Recordings of GluN1/GluN2A receptors with cross-linked GluN2A NTDs revealed that the receptors retain high affinity (nM) zinc binding yet show resistance to zinc inhibition, with a substantially reduced maximal level of zinc inhibition (40% vs 74% for wild-type GluN1/GluN2A receptors; Fig. 4c). In contrast, zinc robustly inhibited GluN1/GluN2B receptors with cross-linked GluN2B NTDs, displaying slightly decreased potency but similar efficacy (i.e., maximal inhibition) compared to wild-type GluN1/GluN2B receptors (Fig. 4d, left panel). Inhibition by ifenprodil remained mostly unperturbed by the GluN2B NTD inter-dimer attachment as evidenced by the closely overlapping dose-responses curves (Fig. 4d, right panel). Hence, maintaining the two NTD heterodimers in close proximity strongly alters allosteric transduction between the NTD layer and the downstream gating machinery in GluN2A receptors but not in GluN2B receptors.

**Distinct NTD-LBD allosteric coupling in GluN2A and GluN2B receptors.** We next investigated the mechanisms coupling the

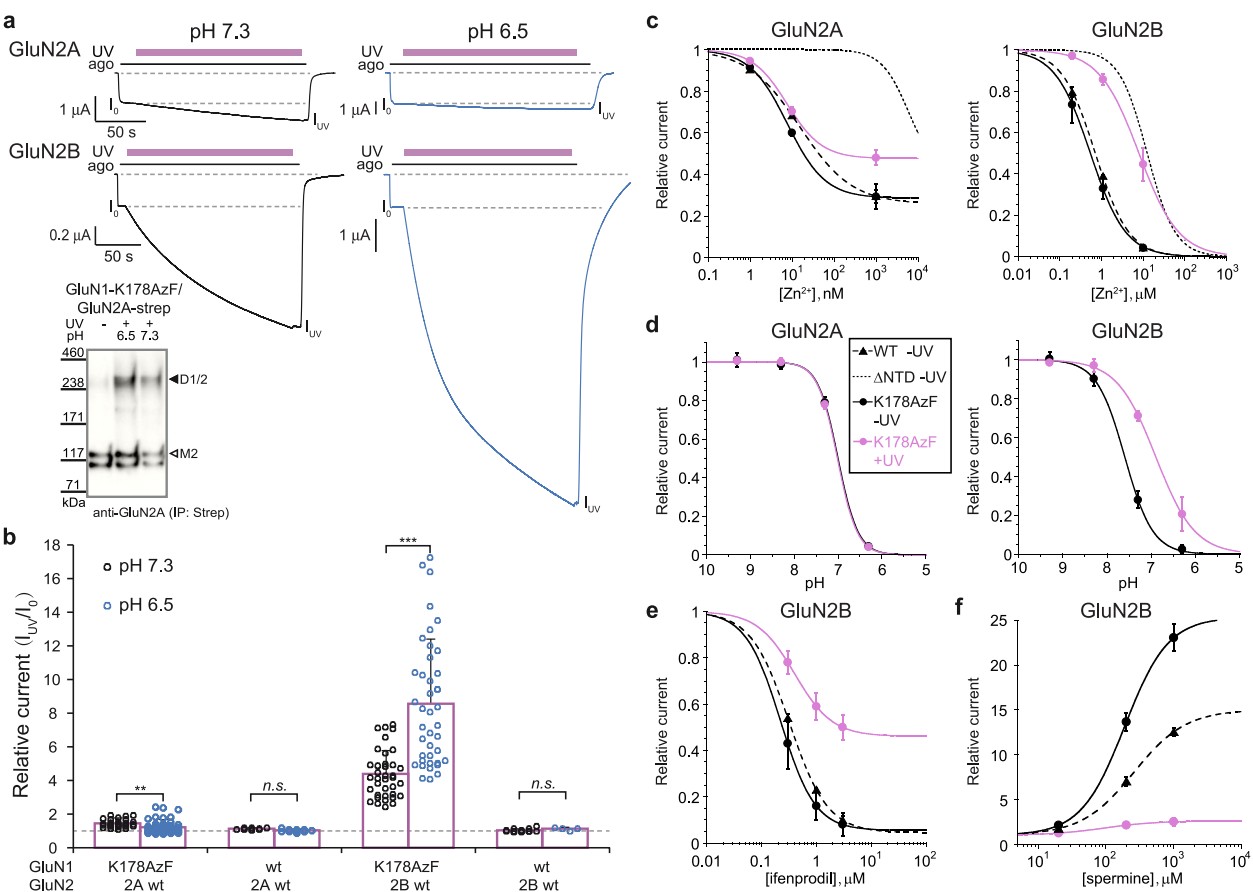

**Fig. 3 Photocrosslinking differentially affects allosteric modulation of GluN2A and GluN2B receptors. a** Representative current traces from oocytes expressing GluN1-K178AzF/GluN2A receptors (top traces) and GluN1-K178/GluN2B receptors (bottom traces) during UV illumination. Experiments were performed at pH 7.3 (left, black) or pH 6.5 (right, blue). ago, for agonists. Inset: immunoblots from HEK cells expressing GluN1-K178AzF/GluN2A-Strep mutant receptors and exposed to UV (+) or not (−) at pH 6.5 or 7.3. GluN2A-Strep monomer (M2) is expected to run at ~130 kDa, and GluN1/GluN2A heterodimer (D1/2) at ~240 kDa. **b** Change in current amplitude upon UV illumination ($I_{UV}/I_o$) of wild-type (wt) receptors and GluN1-K178AzF/GluN2A and GluN1-K178azF/GluN2B mutant receptors, recorded either at pH 7.3 or 6.5. Values are, from left to right: 1.45 ± 0.23 ($n = 32$) and 1.23 ± 0.37 ($n = 58$) for AzF mutant GluN2A receptors; 1.09 ± 0.10 ($n = 10$) and 1.04 ± 0.07 ($n = 12$) for wt GluN2A receptors; 4.39 ± 1.40 ($n = 37$) and 8.59 ± 3.71 ($n = 41$) for AzF mutant GluN2B receptors; 1.04 ± 0.09 ($n = 16$) and 1.14 ± 0.09 ($n = 4$) for wt GluN2B receptors. Data represent mean ± SD. $n$ = number of biologically independent cells. **P = 0.004, ***P < 0.001, n.s. non-significant ($P = 0.22$ for wt GluN2A and 0.06 for wt GluN2B) (one-way ANOVA). **c** Zinc sensitivity of GluN1-K178AzF/GluN2A receptors (left) and GluN1-K178/GluN2B receptors (right), before (plain black line) and after (violet) UV treatment. Dashed line, wt receptors; dotted line, GluN2-ΔNTD receptors[80]. Values of $Zn^{2+}$ IC$_{50}$, maximal inhibition (for GluN2A receptors) and Hill slope ($n_H$) are: 7.43 ± 1.10 nM, 0.70 ± 0.03 and 0.99 ± 0.09 ($n = 4$) before UV, and 7.69 ± 0.17 nM, 0.52 ± 0.04 and 1.05 ± 0.07 ($n = 3$) after UV for GluN1-K178AzF/GluN2A; 15.33 ± 2.99 nM, 0.74 ± 0.05 and 0.70 ± 0.03 ($n = 4$) for wt GluN1/GluN2A; 0.54 ± 0.01 μM and 1.02 ± 0.02 ($n = 4$) before UV, and 7.96 ± 0.16 μM and 0.91 ± 0.02 ($n = 3$) after UV for GluN1-K178AzF/GluN2B; 0.72 ± 0.03 μM and 1.08 ± 0.04 ($n = 4$) for wt GluN1/GluN2B. Data represent mean ± SD. $n$ = number of biologically independent cells. **d** Proton sensitivity. Conditions as in c. Values of pH$_{IC50}$ and Hill slope ($n_H$) are: 7.00 ± 0.02 and 1.91 ± 0.11 ($n = 9$) before UV, and 7.01 ± 0.01 and 1.93 ± 0.07 ($n = 10$) after UV for GluN1-K178AzF/GluN2A; 7.60 ± 0.01 and 1.37 ± 0.05 ($n = 7-9$) before UV; and 6.90 ± 0.02 and 0.99 ± 0.03 ($n = 5-6$) after UV for GluN1-K178AzF/GluN2B. Data represent mean ± SD. $n$ = number of biologically independent cells. **e** Ifenprodil sensitivity of GluN1-K178AzF/GluN2B receptors before (plain black line) and after (violet) UV treatment. Dashed line, wt GluN1/GluN2B receptors. Values of ifenprodil IC$_{50}$, maximal inhibition and Hill slope ($n_H$) are: 0.32 ± 0.02 μM, 0.95 ± 0.02 and 1.22 ± 0.04 ($n = 3$) for wt; 0.24 ± 0.06 μM, 0.98 ± 0.09 and 1.25 ± 0.39 ($n = 4$) before UV; and 0.40 ± 0.12 μM, 0.54 ± 0.06 and 1.25 ± 0.11 ($n = 4$) after UV for GluN1-K178AzF/GluN2B. Data represent mean ± SD. $n$ = number of biologically independent cells. **f** Spermine sensitivity. Conditions as in e. Values of spermine EC$_{50}$, maximal potentiation and $n_H$ are: 174.3 ± 64.5 μM, 11.2 ± 3.3 and 1.26 ± 0.09 ($n = 3-5$) for wt; 214.9 ± 63.2 μM, 26.1 ± 2.5 and 1.29 ± 0.22 ($n = 4$) before UV; and 93.0 ± 22.8 μM, 1.6 ± 0.5 and 1.14 ± 0.16 ($n = 5$) after UV for GluN1-K178AzF/GluN2B. Data represent mean ± SD. $n$ = number of biologically independent cells.

NTDs and LBDs in GluN2A and GluN2B receptors. In both receptor subtypes, the LBD and NTD layers are stacked one on top of the other and operate as a single allosteric unit that governs the receptor channel activity[13,15,16,47]. Yet whether GluN2A and GluN2B receptor subtypes share common NTD-LBD coupling mechanisms remains unclear. To address this question, we imposed structural constraints of the LBD gating ring and studied the influence on allosteric signaling through the NTDs. In a first

set of experiments, we covalently attached neighboring GluN1 and GluN2 LBDs by engineering disulfide bridges at their upper lobe-upper lobe (D1-D1) dimerization interface (GluN1-CC/GluN2-CC receptors, Fig. 5a and Supplementary Fig. 6). This structural modification greatly diminished zinc inhibition of GluN2A receptors (Fig. 5b), confirming previous results[68]. We obtained further evidence for the strong coupling between the NTD zinc site and the D1-D1 LBD interface in GluN2A receptors

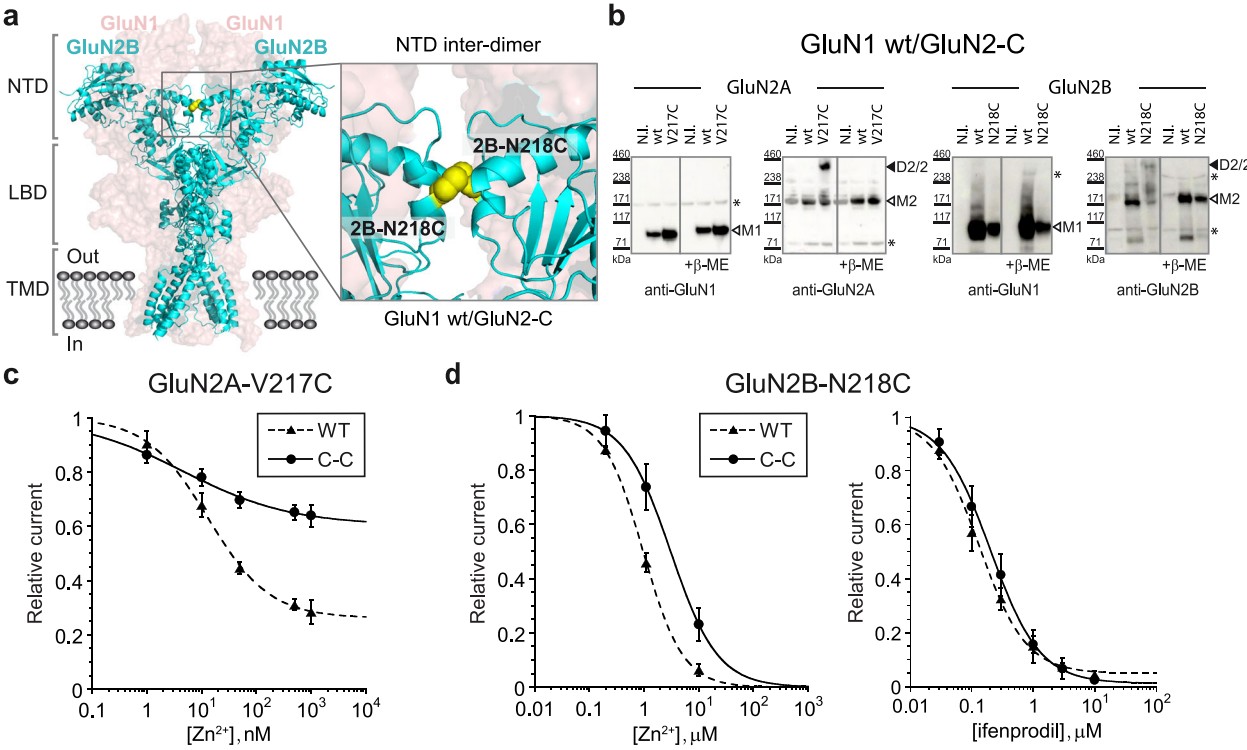

**Fig. 4 Restraining NTD inter-dimer conformational mobility differentially affects GluN2A and GluN2B allostery. a** Structure of the full-length GluN1/ GluN2B receptors (inhibited state PDB 4PE5, Ref. [10]) with the location of the engineered cysteine mutation highlighted (yellow spheres). The two GluN2 subunits are shown in cartoon representation while the two GluN1 subunits in the back are displayed in space-filled. Inset: enlargement of the region with the pair of cysteines introduced to form a disulfide bond at the NTD inter-dimer interface made by the two adjacent GluN2 NTD lower lobes. **b** Immunoblots from *Xenopus* oocytes expressing GluN1/GluN2A-V217C or GluN1/GluN2B-N218C mutant receptors. Samples were analyzed using anti-GluN1 and anti-GluN2A or anti-GluN2B antibodies. GluN1 monomer (M1) runs at ~110 kDa (M1), GluN2A and GluN2B monomer at ~180 kDa (M2), and GluN2 homodimer at ~360 kDa (D2/2). * indicates non-specific background bands. "± β-ME" indicates immunoblots performed with or without β-mercaptoethanol, i.e., in reducing or non-reducing conditions. N.I. for non-injected oocytes. **c** Zinc inhibition dose-response curves of GluN1/GluN2A-V217C receptors (C-C, plain line). For comparison, zinc sensitivity of wild-type GluN1/GluN2A receptors (wt, dashed line) is also shown. Values of $Zn^{2+}$ $IC_{50}$, maximal inhibition and Hill slope ($n_H$) are: 4.64 ± 1.78 nM, 0.40 ± 0.03 and 0.44 ± 0.08 ($n = 7$–13) for C-C; 12.99 ± 1.12 nM, 0.74 ± 0.01 and 0.77 ± 0.05 ($n = 7$–18) for wt. Data represent mean ± SD. $n$ = number of biologically independent cells. **d** Zinc (left) and ifenprodil (right) inhibition dose-response curves of GluN1/GluN2B-N218C receptors (C-C, plain lines). For comparison, zinc and ifenprodil sensitivity of wild-type (wt) GluN1/GluN2B receptors are also shown (dashed lines): Values of $Zn^{2+}$ $IC_{50}$ and Hill slope ($n_H$) are: 3.06 ± 0.04 μM and 1.01 ± 0.01 ($n = 5$) for C-C vs 0.97 ± 0.03 μM and 1.20 ± 0.04 ($n = 4$) for wt. Values of ifenprodil $IC_{50}$, maximal inhibition and Hill slope ($n_H$) are: 0.20 ± 0.01 μM, 0.99 ± 0.02 and 1.08 ± 0.06 ($n = 5$–15) for C-C; 0.13 ± 0.01 μM, 0.95 ± 0.02 and 1.14 ± 0.11 ($n = 9$–16) for wt. Data represent mean ± SD. $n$ = number of biologically independent cells.

by assessing zinc sensitivity in the presence of GNE-3419, a GluN2A-selective positive allosteric modulator (PAM) that binds the D1-D1 interface[32]. GNE-3419 (100 μM) markedly decreased zinc inhibition of GluN2A receptors (Supplementary Fig. 8). It also potentiated these receptors to a much greater extent in the presence of zinc than in its absence (3.01 ± 0.32 [$n = 10$] fold potentiation in 100 nM zinc, a concentration almost fully occupying the high-affinity GluN2A NTD zinc site[69], vs 1.27 ± 0.04 [$n = 9$] without zinc; Fig. 5c). These results are in agreement with the binding of GNE-3419 displacing zinc from GluN2A receptors through long-distant allosteric coupling[50,68] between the NTD zinc site and the D1-D1 LBD intra-dimer interface.

Surprisingly, in the case of GluN2B receptors, no effect was observed on zinc sensitivity when restraining the conformational mobility of the LDB intra-dimer interface, as judged by the superimposed zinc dose-response curves between wild-type and GluN1-CC/GluN2B-CC receptors (Fig. 5d left panel). This lack of functional effect could not be attributed to a deficiency in disulfide bond formation, as verified by Western blots experiments (Supplementary Fig. 9). The D1-D1 crosslink had also modest effect on inhibition by ifenprodil, slightly reducing the extent of maximal inhibition but maintaining its potency

untouched (Fig. 5d right panel). Thus, allosteric inhibition through the NTDs requires LBD intra-dimer conformational mobility in GluN2A, but not GluN2B, receptors. In a final set of experiments, we interrogated the role of the LBD inter-dimer interface on NTD-controlled allosteric routes in GluN2B receptors. Building on our recent finding that a "rolling" motion at an interface between the two constitutive GluN1/GluN2 LBD pairs mediates NTD-LBD coupling in GluN2B receptors[47], we combined cysteine mutations to lock this LBD inter-dimer interface and the photoreactive GluN1-K178AzF mutant to probe for NTD allostery. UV illumination of GluN1-K178AzF-C/ GluN2B-C receptors potentiated receptor activity yet the extent of photo-potentiation was greatly reduced compared to parent GluN1-K178AzF/GluN2B receptors with unconstrained LBDs (1.68 ± 0.56 [$n = 6$] fold potentiation vs 4.95 ± 1.31 [$n = 12$] for control receptors, $P < 0.001$; Fig. 5e, f and Supplementary Fig. 6). In contrast, when pairing K178AzF with the LBD intra-dimer GluN1-CC/GluN2B-CC mutant, photo-potentiation effects were invariably large (5.17 ± 1.18 [$n = 5$]), similar in amplitude to that observed with unconstrained receptors (Fig. 5e, f). For GluN2A receptors, opposite results were observed whereby constraining the LBD intra-dimer, but not inter-dimer, interface, strongly

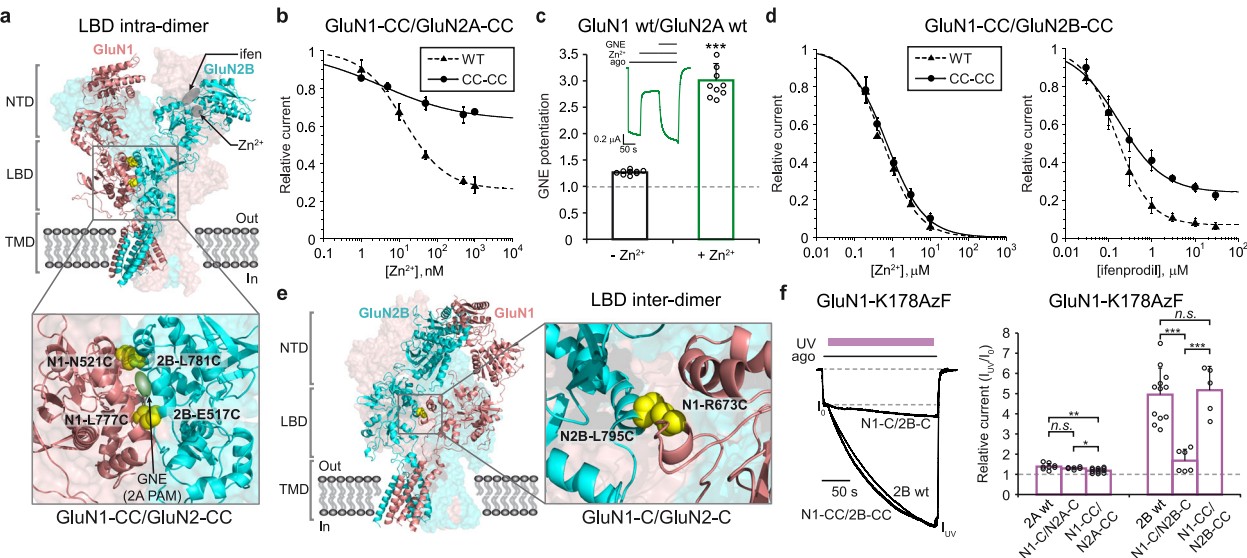

**Fig. 5 Inter-layer NTD-LBD coupling differs between GluN2A and GluN2B receptors. a** Structure of the full-length GluN1/GluN2B receptors (inhibited state PDB 4PE5, Ref. [10]) with the four engineered cysteine mutations at the LBD intra-dimer interface highlighted (yellow spheres). The NTD binding sites for the negative allosteric modulators ifenprodil and zinc are indicated. Inset: enlargement of the region with cysteines introduced. The binding site for GNE-3419, a GluN2A-selective positive allosteric modulator (PAM) is also highlighted (green spot). **b** Zinc sensitivity of GluN1-N521C-L777C/GluN2A-E516C-L780C (CC/CC) receptors (plain line). Dashed line, wild-type (wt) GluN1/GluN2A receptors. Values of $Zn^{2+}$ $IC_{50}$, maximal inhibition and Hill slope ($n_H$) are: $3.44 \pm 2.03$ nM, $0.37 \pm 0.04$ and $0.40 \pm 0.11$ ($n = 4$–14) for CC-CC; $12.99 \pm 1.12$ nM, $0.74 \pm 0.01$ and $0.77 \pm 0.05$ ($n = 7$–18) for wt. Data represent mean ± SD. $n =$ number of biologically independent cells. **c** Potentiation of wt GluN1/GluN2A receptors by GNE-3419 (100 μM) in the absence or presence of the GluN2A NTD inhibitor zinc (100 nM). Values are: $1.27 \pm 0.04$ ($n = 9$) without zinc and $3.01 \pm 0.32$ ($n = 10$) in zinc. Inset: Effect of GNE-3419 in the presence of zinc. GNE-3419 application fully reverses zinc inhibition of GluN2A receptors ago, for agonists. Data represent mean ± SD. $n =$ number of biologically independent cells. ***$P < 0.001$ (two-sided Student's $t$ test). **d** Zinc (left) and ifenprodil (right) inhibition dose-response curves of GluN1-N512C-L777C/GluN2B-E517C-L781C receptors (CC-CC, plain lines). Dashed line, wt GluN1/GluN2B receptors. Values of $Zn^{2+}$ $IC_{50}$ and Hill slope ($n_H$) are: $0.72 \pm 0.05$ μM and $0.88 \pm 0.05$ ($n = 4$) for CC-CC vs $0.59 \pm 0.06$ μM and $0.96 \pm 0.10$ ($n = 2$–3) for wt. Values of ifenprodil $IC_{50}$, maximal inhibition and Hill slope ($n_H$) are: $0.17 \pm 0.03$ μM, $0.76 \pm 0.03$, $n_H = 0.83 \pm 0.13$ ($n = 7$–8) for CC-CC; $0.15 \pm 0.01$ μM, $0.93 \pm 0.01$ and $1.11 \pm 0.07$ ($n = 4$–5) for wt. Data represent mean ± SD. $n =$ number of biologically independent cells. **e** Structure of the full-length GluN1/GluN2B receptors (inhibited state PDB 4PE5, Ref. [10]) with the two engineered cysteine mutations at the LBD inter-dimer interface highlighted (yellow spheres). Inset: enlargement of the region with cysteines. **f** Responsiveness of disulfide-bond linked mutant receptors to UV illumination (365 nm). Left: Representative current traces from oocytes expressing GluN1-K178AzF/GluN2B wt, GluN1-K178AzF-R673C/GluN2B-L795C (N1-C/2B-C) or GluN1-K178AzF-N512C-L777C/GluN2B-E517C-L781C (N1-CC/2B-CC) receptors during UV illumination. ago, for agonists. Right: change in current amplitude upon UV illumination ($I_{uv}/I_o$) of GluN1-K178AzF/GluN2A wt receptors, GluN1-K178AzF/GluN2B wt receptors, and cysteine mutant receptors. Values are: $1.38 \pm 0.15$ ($n = 8$) for GluN1-K178AzF/GluN2A wt; $1.29 \pm 0.06$ ($n = 6$) for N1-C/N2A-C (GluN1-K178AzF-R673C/GluN2A-L794C) and $1.18 \pm 0.11$ ($n = 12$) for N1-CC/2A-CC; $4.95 \pm 1.31$ ($n = 12$) for GluN1-K178AzF/GluN2B wt; $1.68 \pm 0.56$ ($n = 6$) for N1-C/N2B-C and $5.17 \pm 1.18$ ($n = 5$) for N1-CC/2B-CC. Data represent mean ± SD. $n =$ number of biologically independent cells. *$P = 0.034$, **$P = 0.003$, ***$P < 0.001$, n.s. non-significant ($P = 0.21$ for GluN2A and 0.75 for GluN2B) (one-way ANOVA).

affected NTD photo-regulation (Fig. 5f). Overall, these results points to a clear dichotomy between long-distance allosteric communication in GluN2A and GluN2B receptors. In the former, the allosteric route coupling the distal NTDs to the channel gate relies primarily on the LBD intra-dimer interface while in the latter the LBD inter-dimer tetramerization interface is preferentially employed.

**GluN2A and GluN2B receptors sample distinct conformational landscape.** We further investigated allosteric mechanisms of GluN2A and GluN2B receptors using in silico structural analysis and molecular modeling. Exploiting the recent boom of single-particle cryo-EM analysis of full-length GluN2A and GluN2B receptor structures, we first compiled a series of structural parameters relevant for the receptor's allostery. A total of 34 structures in complex with agonists were examined (see "Methods"), grouped in pre-active and inhibited states for GluN2A receptors, and pre-active, non-active and inhibited states for GluN2B receptors. These structures sample multiple states of the receptor' gating cycle, from receptors switched in a pre-open state and ready to gate open their pore ("pre-active"), to receptors bound to agonists but with a closed pore ("non-active"), up to receptors

shifted in long-lived inactive state conformations due to the binding of NTD allosteric inhibitors ("inhibited"). By integrating all the structures into multiple variable analyses, we systematically compared the structural landscape of GluN2A and GluN2B receptors and infer consistency with our functional observations.

Experimentally, the potentiation induced by GluN1-K178AzF NTD photocrosslinking was found to be highly dependent on the level of receptor activity for GluN2B, but not GluN2A, receptors. In the former, the photo-potentiation is inversely correlated with the receptor initial Po, while in the latter it is basically activity-independent (Fig. 3). Inspection of potential inter-subunit crosslinks of GluN1-K178AzF on the various receptor structures reveals a strikingly different pattern between GluN2A and GluN2B receptors. In GluN2A receptors, the majority of "pre-active" "non-active" and "inhibited" structures display numerous potential inter-subunit atomic contacts for photocrosslinking within the GluN1-GluN2 NTD heterodimer (assuming a minimum distance of ~3 Å for crosslink with the AzF nitrene group; Refs. [57,70]). In contrast, in GluN2B, only "pre-active" structures show substantial potential inter-NTD partners (Fig. 6a, b). Investigating NTD configurations provides further evidence for GluN2A and GluN2B conformational divergence. The upper

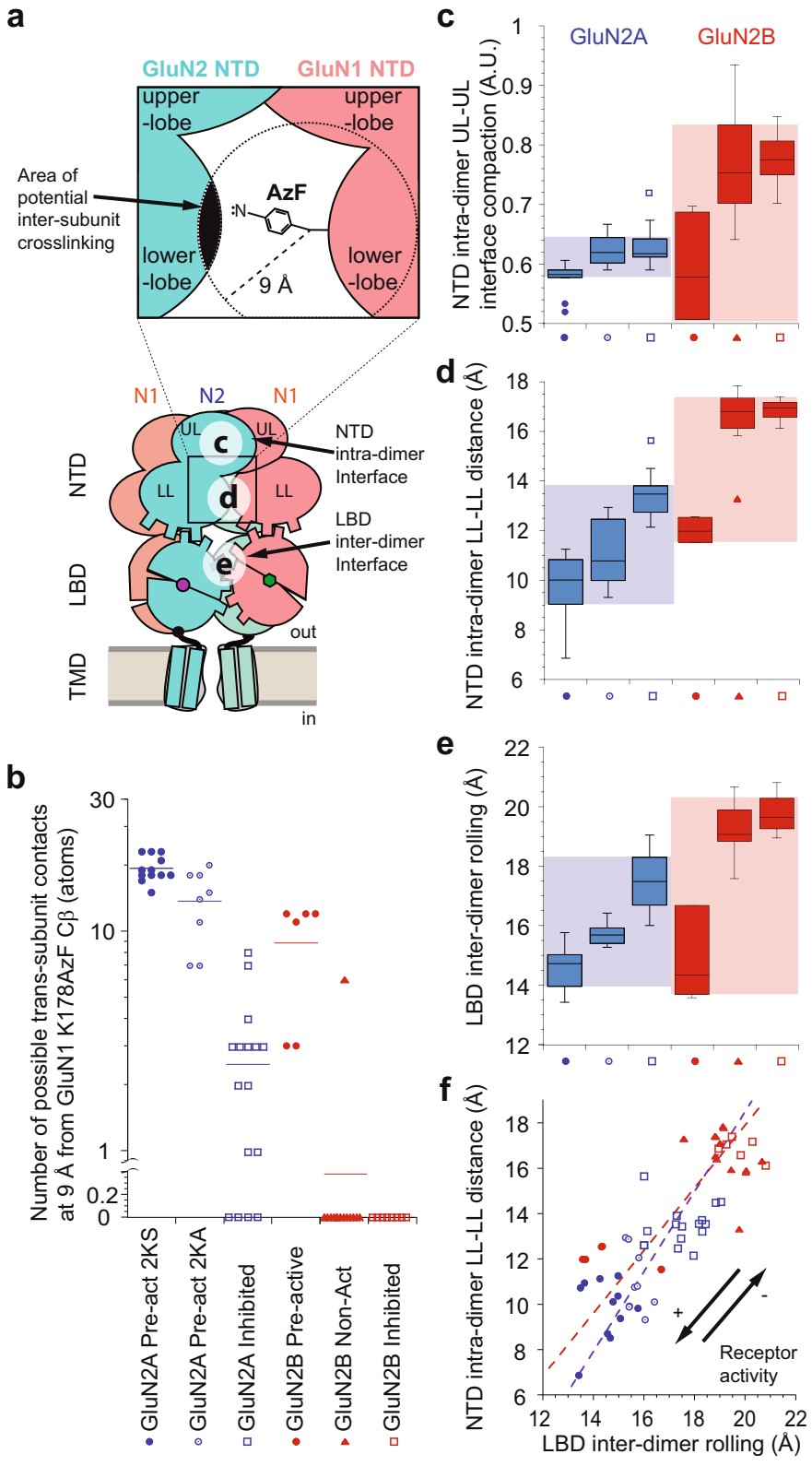

lobe-upper lobe dimer interface, which mediates NTD hetero-dimerization and contains the ifenprodil binding cavity[60], was found to be consistently more compact in GluN2A than GluN2B receptors (Fig. 6c). Most interestingly, the interface cavity is invariably small in GluN2A receptors whatever their functional state, while in GluN2B receptors the cavity fluctuates between small ("pre-active" state) and large ("non-active" and "inhibited"

states) volumes. Similarly, distance measurements between the GluN1 and GluN2 NTD lower lobes show that GluN2B receptors can sample much greater separation than GluN2A receptors (Fig. 6d). Hence, individual NTD heterodimers are more compact and tightly packed in GluN2A than GluN2B receptors. This stable compactness of NTD heterodimers in GluN2A receptors likely accounts for their poor sensitivity to conformational constraints

**Fig. 6 GluN2A and GluN2B receptors exhibit different conformational landscape.** Structural analyses were performed using 29 available full-length NMDAR structures. **a** Lower panel: Localization of the various regions of interest. Letters (**c**, **d** and **e**) indicate the receptor' regions where structural parameters were measured, and refer to plots in corresponding panels (**c**, **d** and **e**). Upper panel: Zoom of the (**d**) region encompassing the lower lobes of the GluN1-GluN2 NTD heterodimer. The estimated region for GluN1-K178AzF photocrosslinking is highlighted as a reaction sphere of 9 Å radius from GluN1-K178AzF Cβ[57,70]. N: reactive nitrene radical, NTD N-terminal domain, LBD ligand-binding domain, TMD transmembrane domain, UL upper lobe, LL lower lobe. **b** GluN2A and GluN2B receptors differ by their accessibility for inter-subunit cross-linking by the photocrosslinker GluN1-K718AzF. The more atoms lie within the 9 Å of AzF Cβ, the closer are the lower lobes within the NTD heterodimer. Horizontal bars represent mean values. Values of n (number of structures) are (from left to right): 12, 8, 16, 6, 16 and 6. **c** The NTD intra-dimer upper lobe-upper lobe (UL-UL) interface adopts various compactions in GluN2B, but not GluN2A, receptors. The smaller the value of compaction, the lower the distance between the two UL protomers. Boxes display interquartile range, median is shown as center line, whiskers extend from the hinge to the largest and smallest value no further than plus (upper whisker) or minus (lower whisker) 1.5 x IQR (interquartile range). Values of n (number of structures) are (from left to right): 12, 8, 16, 6, 16 and 6. **d** The NTD intra-dimer lower lobe-lower lobe (LL) interface displays distinct range of LL separation between GluN2A and GluN2B receptors. Boxes display interquartile range, median is shown as center line, whiskers extend from the hinge to the largest and smallest value no further than plus (upper whisker) or minus (lower whisker) 1.5 x IQR (interquartile range). Values of n (number of structures) are (from left to right): 12, 8, 16, 6, 16 and 6. **e** The range of LBD inter-dimer rolling motion[47] is greater in GluN2B than in GluN2A receptors. Boxes display interquartile range, median is shown as center line, whiskers extend from the hinge to the largest and smallest value no further than plus (upper whisker) or minus (lower whisker) 1.5 x IQR (interquartile range). Values of n (number of structures) are (from left to right): 12, 8, 16, 6, 16 and 6. **f** NTD-LBD conformational coupling is sampled differently in GluN2A and GluN2B receptors. The graph plots the NTD intra-dimer lower lobe-lower lobe (LL) distance as a function the LBD inter-dimer rolling (parameters described in panels d and e, respectively). The dotted lines correspond to linear regressions for all states of GluN2A (blue, R = 0.725) and GluN2B (red, R = 0.729) receptors. Black arrows indicate the change in level of receptor activity (channel Po), increasing (+) or decreasing (−), according to the receptor conformational state.

imposed by GluN1-K178AzF photocrosslinking. At the LBD layer, comparing the rolling motion between the two constitutive LBD dimers[13,47] also indicates greater conformational mobility of GluN2B receptors. In particular, the rolling-down when receptors transit to inactive states ("non-active" and "inhibited") is much more pronounced in GluN2B than GluN2A receptors (Fig. 6e). This distinctive structural behavior of GluN2B receptors is further evidenced in correlation plots between the LBD inter-dimer rolling and the NTD intra-dimer compaction (Fig. 6f), demonstrating that the largest amplitudes of NTD-LBD coupling motions between these two interfaces (NTD intra-dimer and LBD inter-dimer) are encountered solely in GluN2B receptors.

We finally turned to computational modeling to probe the conformational dynamics of NMDARs during allosteric transition. For that purpose, we used iMODFit[71], a structural fitting methodology based on normal mode analysis that has proven useful to study large concerted motions of biomolecules including iGluRs[47,72–74]. Previous modeling on GluN2B receptors clearly pointed to the central role of the rolling motion between the two constitutive LBD dimers in the allosteric cascade connecting the distal NTDs to the channel pore[47]. For GluN2A receptors, we based our modeling on a newly described GluN2A receptor structure captured in the "splayed-open" configuration likely representing a biologically-relevant and long-lived inhibited state bound to allosteric inhibitors (zinc and protons; Ref. [15]). In this structure, not described in similar conditions for GluN2B receptors, the two individual NTD heterodimers conserve their typical fold and high compaction yet are far apart one from another (Fig. 7a; Ref. [15]). Running iMODfit using the full atom model of the "pre-active" 2K state into the density of the "splayed open" state produced a satisfactory fit (mean RMSD dropping from 12.4 to 4.4 Å). The associated trajectory could be decomposed in four consecutive steps, each showing salient features (Fig. 7b, c and see Movie 1). Step 1 involves mostly conformational rearrangements within the NTD layers including twisting motions[56] within individual NTDs; step 2 combines a separation of the two constitutive NTD heterodimers with a large rotation of the TMD; eventually, in the last part of the run (steps 3 and 4), the two NTD dimers separate further apart, in concert with a conspicuous breakage of the LBD intra-dimer D1-D1 interface, a motion expected to relax the tension exerted on the channel gate[75]. Notably, the trajectory as a whole involves

minimal LBD inter-dimer rolling but associates NTD mobility to separation of the LBD intra-dimer interface, in good correlation with our experimental findings on the GluN2A-specific allosteric pathway (see Fig. 5). Overall, this simulation illuminates the importance of subunit-subunit interfaces and their rearrangements in controlling GluN2A NMDAR allosteric dynamics and activity. It also delineates a novel long-distance allosteric route and interlayer conformation spread, distinct from that previously observed at GluN2B receptors.

## Discussion

As main drivers of synaptic plasticity, learning and memory, NMDARs have evolved as particularly complex molecular machines, endowed with exceptional allosteric capacity and ability to sense their extracellular microenvironment[1,35]. Inter-subunit and inter-domain rearrangement is the lynchpin of functional transitions in these multimeric receptors, accounting for receptor channel gating and its allosteric regulation. It is thus essential to improve our understanding of the subunit-specific structural and functional features of NMDARs. In this work, we provide evidence that GluN2A and GluN2B receptors, despite strong structural similarity, proceed through distinct long-range allosteric mechanisms involving different structural routes and rearrangements (Fig. 8 and see below). Our work defines the general principles of GluN2A and GluN2B receptor allosteric mechanisms and provides an integrated understanding of the subunit-specific dynamic regulation of NMDARs. By revealing subunit-specific diversity in NMDAR mechanisms, our work also explains many salient subtype-specific features of NMDARs and offers interesting avenues for new pharmacological interventions.

We introduced the light-sensitive UAAs AzF and BzF at specific sites in the obligatory GluN1 subunit of NMDARs using the genetic code expansion methodology. The UV effects observed at GluN1-K178AzF/GluN2 receptors are remarkable because they can produce very large potentiating effects (on GluN2B receptors; Fig. 1), result in conformational trapping through physical inter-subunit crosslinking (Figs. 1, 3), and display strong subunit dependence (Fig. 3). Although we did not identify the precise crosslinking sites in GluN1-K178AzF/GluN2A and GluN1-K178AzF/GluN2B receptors, the similarity in the photopotentation profile and kinetics of AzF mutants (see Supplementary Fig. 7) as well the strict distance constraints for

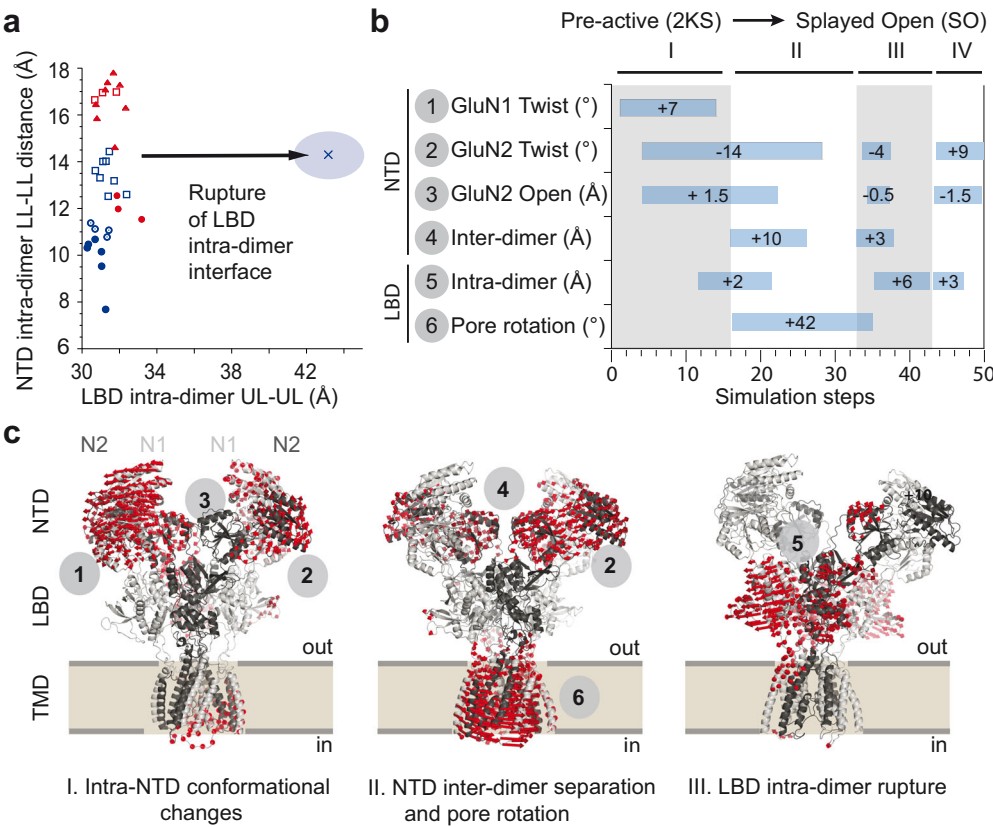

**Fig. 7 Long-distance allosteric coupling in GluN2A receptors. a** The relationship between the NTD intra-dimer distance and the LBD intra-dimer distance as extracted from 29 available full-length X-ray and cryo-EM NMDAR structures (see "Methods"). Same representation as in Fig. 6f, with blue and red colors for GluN2A and GluN2B receptor structures, respectively, and with an additional GluN2A receptor structure captured in a "splayed open" conformation[15] (blue cross). Note that the GluN2A receptor "splayed open" structure displays a disrupted LBD intra-dimer interface but an intact NTD intra-dimer interface. NTD N-terminal domain, LBD ligand-binding domain, UL upper lobe, LL lower lobe. **b, c** iMODfit simulations on a full-length GluN2A receptor from the "pre-active" state to the "splayed-open" state (2KS-6MMR and SO-6MMI, respectively; Ref. [15]). **b** Evolution of selected collective variable during the iMODFit simulation. The trajectory is segmented into four consecutive steps (I to IV). Displacement values greater than 2 Å or 6 degree angle between the starting and the targeted structures are indicated within the corresponding rectangles. **c** Conformational changes experienced by the GluN2A receptor when transiting from the "pre-active" to "splayed-open state" (steps I to III). Red arrows represent protein displacements occurring during each step. Circled numbers refer to the collective parameters analyzed in panel (**b**). Note that during step III, the LBD intra-dimer interface experiences large structural rearrangement eventually leading to its disruption.

photocrosslinking[57,70] point to similar physico-chemical micro-environment for both receptor subtypes, resulting in the trapping of individual NTD dimers in a compact conformation. There are several advantages of using the UAA photo-crosslinker approach to study receptor structure-function relationship. The receptor activity can be assessed before, during, and after the light illumination in a precisely time-controlled manner. It can be combined with the application of various ligands (agonists, antagonists, allosteric modulators), different subunits (in our case GluN2A and GluN2B) and other protein mutations, including, as shown here, alternative disulfide-bond cross-links. Moreover, because photo-crosslinking is strictly distance dependent[57,70], it provides powerful means to study subunit contacts and track state-dependent gating transitions and associated conformational changes[51–54,76]. Exploiting the methodology, we identify distinct GluN1-GluN2 and GluN2-GluN2 interfaces as major conduits for allosteric transduction in NMDARs and show how the functional coupling between these interfaces differs between GluN2A and GluN2B receptors to control receptor function.

Our photocrosslinking experiments show that the local conformation of individual NTD dimers (i.e., the relative arrangement of the GluN1 and GluN2 NTD protomers) has major influence on GluN2B receptor activity, but much less on GluN2A receptors. Our structural analysis also reveals that individual GluN1/GluN2B NTD dimers experience much larger conformational rearrangements than GluN1/GluN2A NTD dimers which adopt a more stable and compact conformation. On the contrary, the respective position of the two adjacent NTD dimers, which form a tetramerization interface between the two GluN2 NTD lower lobes[10,11], strongly affects allostery in GluN2A receptors, but not in GluN2B receptors. In addition, by introducing simultaneous constraints, either physical (crosslinks) or chemical (ligands), at distinct interfaces within the receptor' extracellular region, we establish that GluN2A and GluN2B use distinct inter-layer coupling mechanisms. In GluN2A receptors, NTD-LBD coupling involves the LBD intra-dimer D1-D1 interface while in GluN2B receptors, this later interface plays little role with a major involvement of the LBD inter-dimer interface instead. Our modeling of GluN2A receptor structural dynamics also shows that the LBD intra-dimer D1-D1 interface can break when transiting from the active to the inhibited state, while the same transition in GluN2B receptors[47] favors motions at the LBD

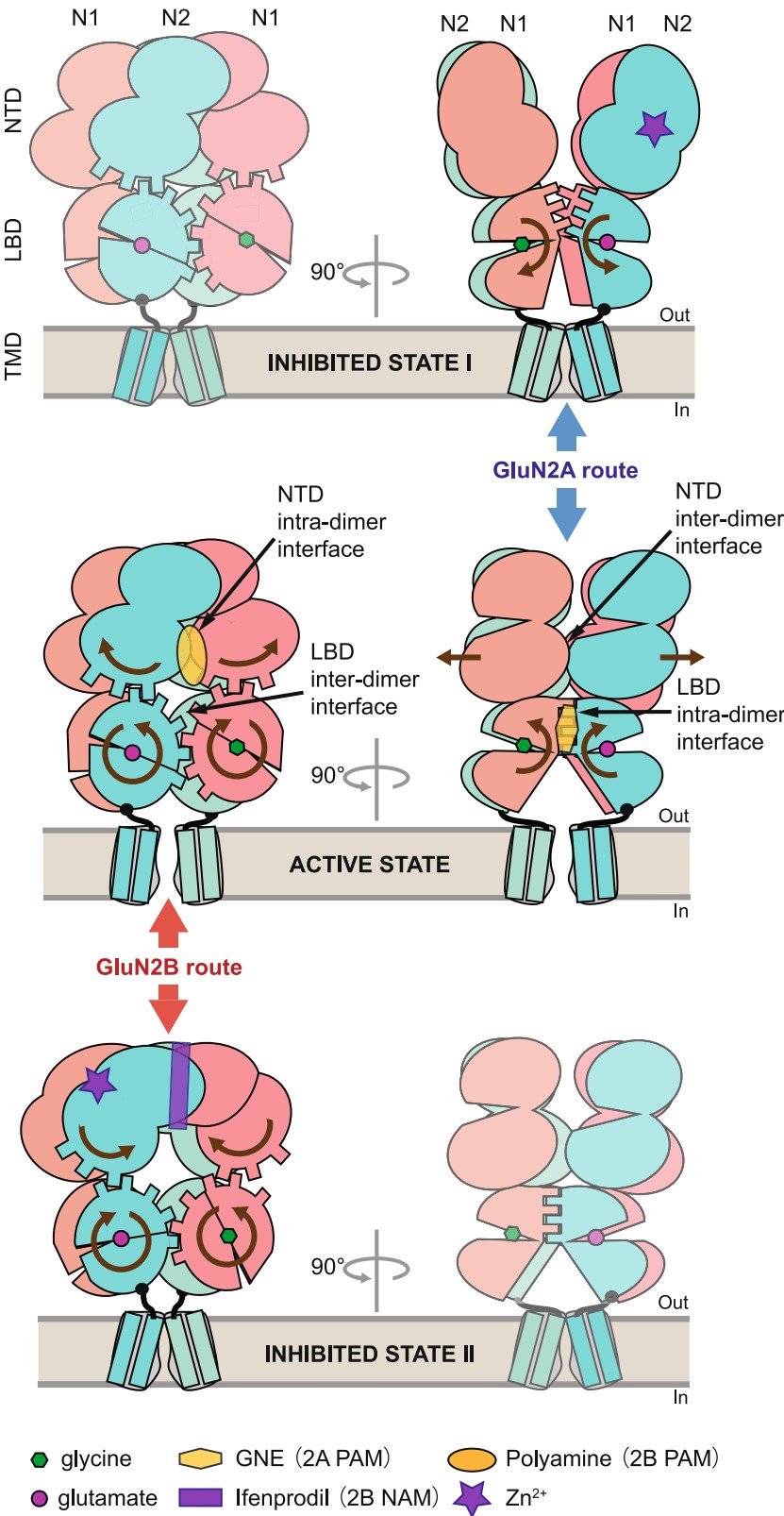

inter-dimer interface with a stable D1-D1 interface. Modeling studies in general reveal that the NMDAR extracellular region explores a large conformational landscape that is exploited to reach specific functional state, depending on subunit composition and interacting ligands[47,72,74,77].

Building on these findings, we propose a comprehensive structural model of GluN2A and GluN2B receptor allosteric regulation via the NTDs (Fig. 8). In this model, the entire NTD-LBD extracellular region acts as a single allosteric unit undergoing large-scale concerted motions that relay to the TMD. Agonist-bound receptors transit between various conformational states associated to an open or closed channel (i.e., active and inhibited states, respectively). Conformational changes occurring at the NTD level are conveyed to the downstream along two possible

**Fig. 8 Proposed mechanisms for inter-layer allosteric transduction in GluN2A and GluN2B receptors.** GluN2A and GluN2B receptors utilize different allosteric routes involving differential subunit–subunit interfaces and conformational rearrangements to couple the NTD layer to the downstream gating machinery (LBD + TMD). Middle and upper rows, GluN2A route. Middle and lower rows, GluN2B route. Binding sites for positive and negative allosteric modulators, NAMs and PAMs, respectively, are indicated. Brown arrows within the receptors indicate motions at play during allosteric transduction (left and right arrows for the GluN2B and GluN2A routes, respectively). Note that the NTD intra-dimer interfaces, both at the level of the upper and lower lobes, are known to form sites for allosteric modulators in GluN2B but not GluN2A receptors. In contrast, the LBD inter-dimer interface is known to form sites for allosteric modulators in GluN2A but not GluN2B receptors. See main text (Discussion) for further details. NTD N-terminal domain, LBD ligand-binding domain also named agonist-binding domain, TMD transmembrane domain.

routes. The "GluN2B route" relies on large conformational rearrangements within individual NTD dimers, that translate into rolling motions between the two LBD dimers[13,47], which in turn modify the tension exerted on TMD segments (Fig. 8, red arrows). The "GluN2A route" on the other hand, relies on motions between the two constitutive NTD dimers, which translate into rearrangements within individual LBD dimers at the level of the D1-D1 interface, a structural element central of channel gate control in all iGluRs[68,78] (Fig. 8, blue arrows). We note that domain swapping (also known as subunit cross-over; Refs.[10,11,45]), such that pairing of domains between the NTD and LBD layers involves different subunits, permits and explains why inter-dimer motions within one layer translate into intra-dimer motions in the neighboring layer and vice versa.

From this dual route model, we infer that zinc, a natural ligand of both GluN2A[50,62,79] and GluN2B[48,80] NTDs, inhibits GluN2A and GluN2B receptors through distinct mechanisms. In GluN2A receptors, zinc binds and closes the GluN2A NTD clamshell[50], which leads to disruption of the NTD inter-dimer interface and separation of the two GluN1-GluN2A NTD heterodimers, a motion that pulls apart the LBD intra-dimer D1-D1 interface, thus relaxing the tension on the LBD-TMD linkers and eventually leading to channel closure (Fig. 8). In such a scheme, interventions that stabilize the D1-D1 interface are expected to oppose zinc inhibition. This is what we observed in the GluN2A PAM GNE-3419 experiment (Fig. 5). In GluN2B receptors, zinc binds and closes the GluN2B NTD[48], which shifts the NTD dimer in a relaxed configuration with the GluN1 and GluN2B lower lobes separating apart, a motion promoting rolling-down of the LBD inter-dimer interface, causing a relief of the tension exerted on the LBD-TMD linkers and eventually channel closure. In such a scenario, compounds that stabilize compact conformation of individual NTD dimers, such as the GluN2B PAM spermine[55], are expected to oppose zinc inhibition through negative allosteric interaction. This is precisely what has been observed[81]. That zinc inhibition engages different allosteric routes in GluN2A and GluN2B receptors also provides a relevant structural framework to explain the differential pH sensitivity of zinc inhibition between the two receptor subtypes[68,79,81], one (GluN2A) displaying high pH dependence but not the other (GluN2B). These differences directly pertain to the complex regulation of neuronal circuits by protons and zinc, two endogenous modulators of NMDARs[1,38,40,41].

Our work further account for key biophysical and pharmacological differences between NMDAR subtypes. Hence, GluN2A receptors have high channel Po[36,64] because their individual GluN1-GluN2A NTD heterodimers are preferentially stabilized in a compact (i.e., active) conformation, acting like build-in PAMs. In contrast, individual GluN1-GluN2B NTDs are much more mobile, undergoing large rearrangements between compact and relaxed (i.e., inhibited) conformation, thus resulting in a lower channel Po[36,64]. Regarding action of drug compounds, the strict dichotomy between GluN2A and GluN2B receptor key interface sites can now be rationalized. At the NTD level, GluN2B allosteric modulators like ifenprodil[60] or spermine[55] that bind

interfaces between the GluN1 and GluN2 NTD protomers, have no equivalent in GluN2A receptors because the GluN1/GluN2A NTD dimer is more rigid and tightly packed (smaller interface cavities). In contrast, the LBD intra-dimer D1-D1 interface is a perfect site for allosteric modulators in GluN2A receptors, such as GNE-3419[32] or TCN-201[82], because it is a labile and functionally sensitive region. Such is not the case of the homologous interface in GluN2B receptors, likely explaining the complete lack of GluN2B pharmacology at this site. Although our model links together many critical aspects of NMDAR function, it certainly represents an oversimplified view of the NMDAR machinery given the highly complex kinetic behavior of NMDAR channels[75]. Moreover, we still do not understand what renders NTD and LBD interfaces so intrinsically different in terms of stability and compactness between GluN2A and GluN2B receptors, particularly at regions that are highly conserved between the two receptor subtypes (like the NTD intra-dimer UL-UL[60] and the LBD intra-dimer D1-D1[82] interfaces; see Supplementary Fig. 1). This is a key question since it determines ultimately which of the two allosteric routes, the GluN2A or GluN2B type, is set in motion. Additional data, both structural and functional, are required to elucidate this point. We suspect however an important role of the NTD-LBD linkers, short connectors divergent in sequence and strategically located between domains and layers[15,16,36]. The potential role of the TMD rotation in the receptor' gating transitions (see Fig. 7), already described in other NMDAR modeling studies[74,77], is yet another point to be clarified. With its imposing extracellular region composed of eight intertwined clamshell-like domains, understanding NMDAR integrated dynamics and subtype specificities remains an important challenge.

Because not all NMDAR subtypes contribute equally to CNS diseases[6], current drug discovery efforts are focused on exploiting the diversity in subunit composition and allosteric sites, with the rationale that subunit-selective modulators can be more effective and better tolerated than the non-selective modulators[31–34]. Sites at dimeric interfaces of GluN1/GluN2 NMDARs have already yielded a rich pharmacology[31,32,82]. Our work now highlights NMDAR tetrameric interfaces (between the two NTD dimers and the two LBD dimers) as promising additional drug sites for subunit-specific fine-tuning of NMDAR signaling and pharmacological interventions against synapse dysfunction.

## Methods
**Plasmids and site-directed mutagenesis**. The pcDNA3-based expression plasmids for rat GluN1-1a (named GluN1 herein), rat GluN2A, and mouse GluN2B (also known as ε2) subunits, Pfu-based site-directed mutagenesis strategy, and sequencing procedure have been described previously[68,69]. Incorporation of the unnatural amino acids (UAAs) AzF and BzF into NMDARs was performed using the genetic code expansion methodology as previously described[52,53,83]. The plasmid pSVB.Yam carrying the gene encoding the amber suppressor tRNA derived from B. *stearothermophilus* Tyr-tRNA_{CUA}, while the aminoacyl-tRNA synthetases (aaRS) construction for AzF and BzF derived from *E. coli* TyrRS[84,85].

**Two-electrode voltage clamp (TEVC) recordings**. Recombinant NMDARs were expressed in *Xenopus laevis* oocytes after nuclear injection of 36 nl of a mixture of

cDNAs (at 10–30 ng/μl) encoding various GluN1 and GluN2 subunits (ratio 1:1). Oocytes were prepared, injected, incubated, and voltage-clamped as described previously[68]. For UAA incorporation, oocytes were co-injected with 36 nl of a cDNA mixture encoding for of GluN1, GluN2, Yam, and aaRS as follows, unless otherwise indicated in the text: GluN1-K178amber (120 ng/μl), GluN2A (80 ng/μl) or GluN2B (120 ng/μl), Yam (10 ng/μl), AzF-RS (5 ng/μl). After injection, oocytes were incubated at 19 °C in a Barth solution containing (in mM) 88 NaCl, 1 KCl, 0.33 Ca(NO$_3$)$_2$, 0.41 CaCl$_2$, 0.82 MgSO$_4$, 2.4 NaHCO$_3$, 10 HEPES (pH adjusted to 7.6 with NaOH) supplemented with gentamicin (50 ng/ml) and the NMDAR glutamate binding site competitive antagonist D-APV (50 μM). AzF and BzF were added to the incubation medium at final concentrations of 1 mM and 2 mM, respectively.

For all experiments, except those on pH and spermine sensitivity, the standard external solution contained (in mM): 100 NaCl, 0.3 BaCl$_2$, and 5 HEPES (pH adjusted to 7.3 with KOH). Moreover, in all "0" zinc solutions, 10 μM DTPA was added to chelate contaminating zinc and prevent tonic NMDAR inhibition[69]. For zinc DRC on GluN1/GluN2A receptors, tricine (10 mM) was used to buffer zinc[69], and the following relationship was used to calculate the free zinc concentrations[61,68]: [Zn]$_{free}$ = [Zn]$_{added}$ / 200. For zinc DRC on GluN1/GluN2B receptors, no zinc buffer was used and zinc concentrations were corrected for an estimated contaminant zinc concentration of 100 nM (ref. [80]). For the pH and spermine sensitivity experiments, an external solution enriched in HEPES was used to insure proper pH buffering[68]. Unless otherwise mentioned, NMDAR-mediated currents were induced by simultaneously applications of saturating concentrations of glutamate and glycine (100 μM each). Glutamate and glycine dose-response curves (DRC) were performed in the presence of 100 μM glycine and 100 μM glutamate, respectively. Recordings were performed at a holding potential of −60 mV and at room temperature. Data were collected and analyzed using Clampex 10.5 and Clampfit 10.5 (Molecular Devices), respectively, and fitted using KaleidaGraph 4.0 (Synergy software).

Agonist DRC were fitted with the following Hill equation: I$_{rel}$ = 1 / (1 + (EC$_{50}$/ [A])$^{nH}$), where I$_{rel}$ is the mean relative current, [A] the agonist concentration, and nH the Hill coefficient. EC$_{50}$ and nH were fitted as free parameters. Ifenprodil and zinc DRC were fitted with the following Hill equation: I$_{antago}$/I$_{control}$ = 1 − a / (+ (IC$_{50}$/[B])$^{nH}$), where I$_{antago}$/I$_{control}$ is the mean relative current, [B] the ifenprodil or free zinc concentration, (1 − a) the maximal inhibition, and nH the Hill coefficient. IC$_{50}$, a, and nH were fitted as free parameters except when fitting zinc sensitivity of GluN1/GluN2B receptors (a fixed to 1). For proton DRC, corrections for small shifts in the reference potential of bath electrodes and analysis of the data were performed as previously described[68]. Spermine experiments were performed and analyzed as previously described[55]. In brief, spermine sensitivity was assessed at pH 6.5 and at a holding potential of −30 mV to maximize the spermine induced potentiation. Spermine DRC were fitted using the following Hill equation: I$_{spermine}$/ I$_0$ = 1 + a / (1 + (EC$_{50}$/[spermine])$^{nH}$), where I$_{spermine}$/I$_0$ is the mean relative current, [spermine] is the spermine concentration, nH is the Hill coefficient, and (a + 1) the maximal potentiation at saturating spermine concentrations. EC$_{50}$, a and nH were set as free parameters. MK-801 is an NMDAR open channel blocker with slow reversibility and, consequently, the rate at which MK-801 inhibits NMDAR responses depends on the level of channel activity, that is, channel open probability (Po). Based on this principle, MK-801 is classically used to estimate the level of NMDAR channel activity in macroscopic whole-cell recordings[36,47,56,86,87]. Note, however, that MK-801 inhibition kinetics can index relative, not absolute, Po. MK-801 solutions (10 nM for GluN2A-containing receptors and 50 nM for GluN2B-containing receptors) were prepared by dilution of a stock solution (50 mM) into the agonist containing solutions. MK-801 time constants of inhibition (τ$_{on}$) were obtained by fitting currents with a single-exponential component to a time window corresponding to 10–90% of maximum inhibition. On-rate (k$_{on}$) constants were then calculated assuming a pseudo first-order reaction scheme: k$_{on}$ = 1 / ([MK-801]*τ$_{on}$). Each constant was then normalized to the mean k$_{on}$ constant of the corresponding wild-type receptors measured in the same conditions on the same day. MTS compounds were prepared and used as previously described[36,56]. The time course of current photo-potentiation upon UV illumination was analyzed using mono-exponential fits (time constant τ$_{UV}$). We noted that for some cells, UV-induced photopotentiation showed no apparent saturation over the time course of the experiment, rendering quantification of the kinetics difficult if not impossible. Because it is know that long periods (minutes) of NMDAR activation expressed in Xenopus oocytes—as it is the case here—can lead to slow activation of endogenous oocyte conductance[88], we purposely excluded these cells from the kinetics analysis. To promote formation of disulfide bonds, oocytes were incubated for at least 10 min in DTNB (0.5 mM) in Barth solution supplemented with DTPA (10 μM).

### Whole-cell and outside-out patch-clamp recordings.

Wild-type and mutant NMDARs were expressed in HEK-293 cells (obtained from ECACC, Cat #96121229). HEK cells were cultured in DMEM medium supplemented with 10% fetal bovine calf serum and 1% Penicillin/streptomycin (5000 U/ml). Transfections were performed using polyethylenimine (PEI) in a cDNA/PEI ratio of 1:3 (v/v). Cells were co-transfected with a DNA-mixture containing plasmids encoding wild-type GluN1 or mutant GluN1-K178amb, wild-type GluN2A or GluN2B, tRNA-AzFRS[52], and eGFP. Typically, the total amount of DNA was 1.0 μg per 0.8 cm

cover slip and the mass ratio was 2:1:1:1. AzF (1 mM) and D-APV (150 μM) were added to the culture medium immediately after transfection. Patch-clamp recordings, either in the whole-cell (macroscopic current recordings) or outside-out (single-channel recordings) configuration were performed 48 h following transfection. The extracellular solution contained (in mM): 140 NaCl, 2.8 KCl, 1 CaCl$_2$, 10 HEPES, 20 sucrose and 0.01 DTPA (290–300 mOsm), pH adjusted to 7.3 using NaOH. Patch pipettes had a resistance of 3–6 MΩ (whole-cell) or ~10 MΩ (outside-out) and were filled with a solution containing (in mM): 115 CsF, 10 CsCl, 10 HEPES and 10 BAPTA (280–290 mOsm), pH adjusted to 7.2 using CsOH. Currents were sampled at 10 kHz and low-pass filtered at 2 kHz (except for noise analysis experiments, filter set at 5 kHz) using an Axopatch 200B amplifier and Clampex 10.6 (Molecular Devices). Agonists (100 μM glutamate and 100 μM glycine) were applied using a multi-barrel solution exchanger (RSC 200; BioLogic). Recordings were performed at a holding potential of −60 mV and at room temperature.

Whole-cell noise analysis was performed from segments of at least 10 s of steady currents in the dark and under (or after) UV illumination (≥10 s each, same cell). The channel open probability (Po) was estimated from the whole-cell current noise using the following equation[89]: Po = 1 − var(I) / (i$_{el}$ x m$_I$), where m$_I$ is the mean current amplitude evoked by the agonists, var(I) the variance of the current around the mean, and i$_{el}$ the amplitude of the unitary current as measured from all-points amplitude histograms obtained from independent outside-out single-channel recordings performed in the exact same conditions and lasting at least 4 s (up to 70 s).

### Photo-crosslinking.

For all recordings in *Xenopus* oocytes, UV treatment was performed "on-line" with a 365 nm LED from a PE-2 light source (CoolLED). The light was directly applied to the dark hemisphere of the oocyte in the recording chamber through an optical fiber. The total power measured at a distance of 200 mm from the source is 105 mW (42 mW/cm$^2$), as reported by the manufacturer. The duration of UV applications was usually ~3 min (unless otherwise mentioned). For recordings on HEK cells, UV light was generated using a Mic-LED-365 light source (200 mW, Prizmatix) and delivered via the microscope optical port through a 10x objective. The duration of UV applications was 30–40 sec. For western blot experiments, HEK cells expressing AzF mutant receptors were irradiated for 30 min with a UVP Dual Tube hand-held UV lamp (365 nm, 6 W, Fisher Scientific) placed on top of the plate, the latter being kept on ice. During irradiation, cells were incubated in PBS pH 7.3 or pH 6.5 as indicated. After UV treatment, cells were subject to the immunoblotting assays.

### Immunoblotting.

For cysteine mutant receptors, oocytes were cultured for at least three days post injection to achieve high receptor expression. Sixteen oocytes were selected based on their high level of NMDAR expression as assessed by TEVC recordings. Selected oocytes were homogenized on ice by back and forth pipetting with 160 μl of lysis buffer (20 mM Tris pH 8.0, 50 mM NaCl, 1% DDM, 1/10 of a complete protease inhibitor cocktail tablet, Roche Complete, Mini) until a homogenous suspension was obtained. Samples were then centrifuged at 20,000 g for 5 min at 4 °C, re-homogenized, and centrifuged again. Supernatants enriched in membrane proteins were collected and separated in two equal volumes for subsequent Western blotting experiments in non-reducing and reducing (10% v/v β-mercaptoethanol added to the loading buffer) conditions. Samples were migrated on SDS-PAGE gradient gels (3–8%), dry transferred to PVDF membranes, and finally immunoblotted with the corresponding antibodies.

For AzF mutant receptors, HEK cells, rather than Xenopus oocytes, were preferentially used to obtain sufficient protein amounts. The GluN2 constructs used for HEK transfection were BacMam GluN2B-CTD truncated-mRuby-strep and GluN2A-CTD truncated-GFP-strep (generous gift from Dr. Shujia Zhu). Transfected cells were first collected by centrifugation at 1000 g for 5 min at 4 °C. Cells were then lysed by sonication and centrifuge at 7000 g for 20 min at 4 °C. The supernatant was collected and ultracentrifuged at 40,000 g for 1 hr at 4 °C. The pellet was collected and resuspended with TBS containing 150 mM NaCl, 20 mM Tris and protease inhibitor cocktail (pH adjusted to 7.8). Membranes were then solubilized with a buffer containing 150 mM NaCl, 20 mM Tris, 2% MNG-3, 2 mM glutamate, 2 mM glycine (pH adjusted to 7.8) for 1.5 hr at 4 °C, then ultracentrifuged at 40,000 g for 1 hr at 4 °C. The supernatant was collected and incubated with Strep-tag resin for 30 min at 4 °C. The mixture was centrifuged at 1000 g for 2 min and washed twice in a solution containing 150 mM NaCl, 20 mM HEPES, 0.1 % MNG-3 (pH 7.8). The resin was eluted with a buffer containing 150 mM NaCl, 20 mM HEPES, 0.1 % MNG-3, 5 mM D-Desthiobiotin (pH adjusted to 7.8), rotated for 30 min, and centrifuged at 1000 g for 2 min. The sample was mixed with SDS-loading buffer and boiled at 95 °C for 5 min. SDS-PAGE and immunoblotting were then performed as described above.

The following antibodies were used: anti-GluN1 (1:750, mouse monoclonal MAB1586 clone R1JHL; Millipore, Billerica, MA), anti-GluN2A (1:500, rabbit monoclonal 04-901 clone A12W; Millipore), anti-GluN2B antibody (1:500, mouse monoclonal 75–101 clone N59/36; NeuroMab, UC Davis, CA) and anti-Strep (1:1000, rabbit monoclonal ab180957 clone EPR12666, Abcam, Cambridge, UK). Protein bands were visualized by using secondary goat peroxidase-conjugated anti-mouse antibody (1:20,000, Jackson ImmunoResearch, West Grove, PA, USA) with SuperSignal West Pico Chemiluminescent Substrate (Thermo Scientific, Waltham,

MA, USA). For each targeted position, crosslinking formation was assessed from at least two independent Western blot experiments using different batches of cells.

**Chemicals**. Salts, L-glutamate, glycine, DTPA (diethylenetriamine-pentaacetic acid), tricine (N-tris(hydroxymethyl)methylglycine), DTNB (5,5'-dithio-bis(2-nitrobenzoïc acid)) and spermine were purchased from Sigma-Aldrich (St. Louis, MO, USA), D-APV (D-(-)-2-Amino-5-phosphonopentanoic acid) from Ascent Scientific (Weston-Super-Mare, UK) and gentamycin from GIBCO (Invitrogen, Rockville, MD, USA), D-APV and (+)-MK-801 from Ascent Scientific (Bristol, UK). The thiol-modifying reagents 2-aminoethylmethanethiosulphonatehydrobromide (MTSEA) and 2-[tri-methylammonium]ethylmethanethiosulphonatebromide (MTSET) were obtained from Toronto Research Chemicals (North York, ON, Canada), MNG-3 (maltose neopentyl glycol-3) from Anatrace (Maumee, OH, USA). L-glutamate and glycine stock solutions (100 mM to 1 M) were prepared in bidistilled water. DTPA was prepared as 10 mM stock aliquots at pH 7.3. DTNB was prepared as a stock solution (50 mM) in an HEPES solution (500 mM, pH 8.0). Spermine was directly prepared from powder into agonist solution, freshly every day of experiment. Zinc was prepared as a 100 mM $ZnCl_2$ stock solution (in 1% HCl). Ifenprodil (generous gift from Synthélabo, France) was prepared as 10 mM stock aliquots (in 1% HCl). MK-801 was prepared as 50 µM stock aliquots. p-azido-L-phenylalanine (AzF) was purchased from Chem-Impex International (Wood Dale, IL, USA) and prepared as 10 mM stock solutions in Barth solution with sonication. 4-Benzoyl-L-phenylalanine (BzF; also known as Bpa) was purchased from Bachem (Budendorf, Switzerland) and prepared as 260 mM stock solutions in $ddH_2O$ by equal molar addition of NaOH.

**Structural analysis, and molecular modeling and fitting**. Except for Fig. 7, all structural illustrations of full-length NMDARs are snapshots from a modeled structure of the GluN1/GluN2B receptor captured in an inhibited state as described[47]. The structural analysis presented in Fig. 6 was performed using a large structural dataset including most published full-length X-ray and cryo-EM NMDAR structures (Supplementary Table 1; all GluN2A structural states classified and named according to Ref. [15]): six structures of GluN2A receptors in the pre-active state 2KS (2-knuckle symmetric)[15]; four structures of GluN2A receptors in the pre-active state 2KA (2-knuckle asymmetric)[15]; eight structures of GluN2A receptors in an inhibited state (described as "extended" and "1 K")[15]; three structures of GluN2B receptors in the pre-active state[13,17]; eight structures of GluN2B receptors in the non-active state[2,13,17]; and three structures of GluN2B receptors in an inhibited state[10,11]. The various states of the receptor are defines as follows: "pre-active" for agonist-bound, compact NTD dimers (linked to high Po mode) but with the pore closed; "non-active" stands for agonist-bound, relaxed unliganded NTD dimers (linked to low Po mode). "Inhibited" for agonist-bound, relaxed NTD dimers bound to ifenprodil or Ro-256981, GluN2B-selective allosteric inhibitors (linked to low Po mode). Note that because of the internal two-fold symmetry within the receptor structures, two sets of parameters were extracted per structure. Please also note that no structure in complex with competitive antagonists was included in the analysis. Whether these structures represent native conformations or not remains debated[90]. Rather, all 34 selected structures are in complex with agonists, to focus on the transition between the active and allosterically inhibited (ligand-bound) states.

The GluN1/GluN2 NTD intra-dimer upper lobe-upper lobe interface compaction (Fig. 6c) was estimated by counting the number of GluN1 atoms (Nba) present at <16 Å from the Cα of GluN2B-Q110 (or GluN2A-Q111) in all the selected structures (for subunit pairs A-D and B-C). This latter residue lies at the center of the upper lobe-upper lobe interface[91]. The compaction is expressed as 100/Nba. The NTD intra-dimer lower lobe-lower lobe separation (Fig. 6d) was calculated by measuring the distance from Cα of GluN1-K178 to the Cα of GluN2B-N184 (N179 in some structures) or to the homologous GluN2A-S185 (for subunit pairs A-D and B-C). These two residues face each other at the surface of the NTD lower lobes. The distance thus measures a combination of spacing and rotation of the lower lobes within the NTD dimer. The rotation component is often neglected looking at distances between center-of-mass although it is known to be critical for receptor function[56]. The "rolling" between the two constitute LBD heterodimers[13,47] was estimated by measuring the difference of distances between the centers of the LBD upper lobes (ULs) and the center of the LBD lower lobes (LLs) at the LBD inter-dimer rolling interface (for subunits A-B and C-D). Hence, rolling (A-B) = (Distance center-UL_A to center-UL_B) − (Distance center-LL_A to center LL_B). The distance between the LBD intra-dimer ULs (D1-D1 interface) corresponds to the mean distance between the centers of mass of the LBD ULs of subunits A-D and B-C. For iMODFit simulations, the structural model of the full-length GluN1/GluN2A receptor (CTD excluded) was generated using Modeler, combining available structural data and a "loop-model" reconstruction module to build missing loops and linkers. The model of the 2KS state (PDB 6MMR) was then fitted into the density of the splayed-open state (EM-9152) using iMoDfit[71] and following the procedure described in Ref. [47]. The collective variables used to analyze the iMODfit trajectories are the following: GluN1 and GluN2 NTD interlobe twist angle, defined as in Ref. [56]; GluN2 NTD opening, measured as the distance between the Cα of GluN2A L98 and I256; "NTD inter-dimer distance" measured between Cα of GluN2A K214 (subunit B) and L248 (subunit D); "LBD intra-dimer distance" corresponding to the spacing between LBD ULs (D1-D1 interface) and measured between the Cα of GluN2A A508 and GluN1 I513; "Pore

rotation" measured as a dihedral angle between the Cα of GluN1 (subunit C) Ala 717 – Pro 670 – Gly 620 – I835. The structural illustrations were prepared with PyMOL (http://pymol.org).

**Statistical analysis and reproducibility**. Except for box plots, data are presented as mean ± standard deviation of the mean (SD). Unpaired Student's t-test or one-way ANOVA was used to assess statistical significance. All sample numbers (n) and statistical tests are specified in the text and figure legends. Statistical significance are indicated with *, ** and ***, when P values are below 0.05, 0.01 and 0.001, respectively. n.s. indicates non-significant. Significance was defined as P < 0.05. All immunoblotting experiments were repeated at least two times with similar results.

**Reporting summary**. Further information on research design is available in the Nature Research Reporting Summary linked to this article.

## Data availability
The data that support this study are available from the corresponding authors upon reasonable request. Previously reported structural data used in this study are available from the protein data bank under accession codes 6MMR and 4PE5. Source data are provided with this paper.

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

## Acknowledgements

This work was supported by the French government ("Investissements d'Avenir" ANR-10-LABX-54 MEMOLIFE, ANR-11-IDEX-0001-02 PSL Research University, ANR-11-LABX-0011-01), the European Research Council (ERC Advanced Grant #693021 to PP), the French Academy of Sciences (Prix Lamonica to PP; fellowship to MT), the Chinese Scholars Council (CSC fellowship to MT) and Sorbonne Université (fellowship to LP). We thank Shujia Zhu (Chinese Academy of Sciences, Institute of Neuroscience, Shanghai) for plasmids of strep-tagged CTD-deleted GluN2 constructs (Western blots on HEK cells) and Laetitia Mony (IBENS, Paris) for comments on the paper.

## Author contributions

M.T. performed all the experiments including some of the electrophysiological recordings on HEK cells. D.S. performed all the structural analysis and modeling. L.P. performed electrophysiology experiments on HEK cells. M.D. provided technical help with molecular biology and biochemistry. P.P. and S.Y. designed the study, analyzed the data and supervised the project. M.T, D.S, S.Y. and P.P. wrote the paper.

## Competing interests

The authors declare no competing interests.
