## [Peer Review File · Nature Communications]

Reviewers' Comments:

Reviewer #1:

Remarks to the Author:

The NMDA receptor is a subtype of ionotropic glutamate receptors that plays a key role in synaptic plasticity. Understanding NMDA receptor function and allosteric modulation is of great importance. Tian et al. use two-electrode voltage clamp in combination with unnatural amino acid-based photo-crosslinking and cysteine crosslinking to discover that GluN1/GluN2A and GluN1/GluN2B diheteromeric NMDA receptors take different conformational routes when gating is altered by negative and positive allosteric modulators. These findings show that stabilization in distinct conformations can achieve the same functional regulation. The manuscript is well written. The experiments are done carefully with appropriate controls and presented clearly. The findings are novel and advance the field substantially.

Two general points and several minor points should be addressed.

General Points

- 1) When comparing GluN2A and GluN2B receptor structures, the authors should take into consideration differences in stabilizing mutations, linker deletions and other sequence modifications that can affect the observed differences between GluN2A and GluN2B receptors structures. They should also comment on differences due to potential artifacts such as receptor instability in cryo-EM studies (Mayer 2017; e.g. the discrepancy in results of Chou 2020 and Jalali-Yazdi 2018 for the splayed-open NTD configuration in presence of competitive antagonist or Jalali-Yazdi 2018 vs Zhang 2018 for pH effect on NTD.)
- 2) The authors should compare their iMODfit simulations with published MD simulations, e.g. opening and closing of the channel from Cerny (2019, 10.3390/biom9100546)

Minor points

- 1) The authors should include all of the PDB identification codes in the text (and/or in a supplemental table) to make it clear to which structures they refer.
- 2) I am aware that the NMDA constructs used in x-ray crystallography and cryo-EM are often called "full-length", however I would rather suggest to use e.g. "near full-length" throughout the manuscript since these receptors lack the whole C-terminal domain.
- 3) Figure Legends: Missing Hill slope error value. Is there any reason for this?
- 4) Line 136: ">17 Å", is this Ca distance?
- 5) Something funny happened to several parts of this sentence:
Line 49-52: "One released in the synaptic cleft, glutamate mediates fast neurotransmission by acting on ionotropic glutamate receptors (iGluRs) are subdivided into three main families. AMPA, kainate and NMDA receptors. While AMPA and kainate receptors are primarily involved into glutamate depolarization of the postsynaptic membrane"
- 6) Line 58: "(GluN2A-D)"
- 7) Line 299: " receptor "
- 8) Line 257: 7.0 ± 0.02 ; Line 261: 7.6 ± 0.01 and similarly throughout the text.

Reviewer #2:

Remarks to the Author:

In this interesting paper Paoletti and co-workers assess allosteric signalling in GluN2A vs GluN2B

NMDAR heteromers. Using cross-linking approaches at strategic sites combined with patch clamp recordings they convincingly show differences in allosteric transmission from the distal NTD to the ion channel. Despite their overall structural similarity GluN2A and GluN2B NTD heterodimers show distinct dynamics which are transmitted differently through the LBD layer down to the ion channel. They also explore the rich pharmacology of these iGluRs to further our understanding of allosteric regulation of NMDAR by zinc and protons and differential regulation of the 2 NMDAR subtypes by these agents. This is a nicely conducted study that will be of interest to the wider community of neuroscientists, biochemists and structural biologists. The paper is clearly written and the figures document the story well.

I have some additional comments.

1. The inter-LBD rolling, which seems a crucial point for GluN2B receptors (Fig. 5e-f) has not been tested for GluN2A heteromers. The latter are sensitive to intra-LBD dimer x-links, which seem to be ineffective in GluN2B receptors. Has an inter-LBD x-link been attempted in NR2As? This allosteric point comes up again in the modelling section at the end of the paper (Fig. 6), and would be good to include.
2. could the authors comment on the impact of the GluN1 (exon5) splice variant in the allosteric routes they describe, ie does it behave the same or differently?
3. Supplementary Fig 2b also shows E185AzF, which is not mentioned. This mutant is shown in Fig 1c and not being potentiated by UV light and therefore it may make sense to remove it from Supplementary Fig 2b?
4. line 193: NTD dynamics have also been described in FRET studies (Sirrieh R et al., JBC 2013) and in simulations (Dutta A. et al., Structure 2012), which should be mentioned here.
5. Line 183: Supplementary Fig. 4 includes GluN2A and GluN2B but this section is only about GluN2B ?

Reviewer #3:

Remarks to the Author:

The work performed by Tian et al. employs unnatural amino acid crosslinking, Cys disulphide engineering, receptor pharmacology and in silico approaches to study the allosteric pathways employed by GluN2A and GluN2B subunits. The experiments are relatively simple, well designed and carefully executed, although they are built largely on repeating previously published mutations and experimental approaches. Based on their findings, the authors conclude that NTD-driven transduction pathways differ between GluN1/2A and GluN1/2B receptors.

Insight on subunit-specific functional coupling between NTD and LBD is of potentially great interest to elucidate how modulators differentially affect GluN1/2A and GluN1/2B receptors. However, the findings of the present work are suggestive, but not conclusive with regards to the key point the authors are trying to make.

Major concerns:

- i) The authors outline the strengths of their crosslinking approach and convincingly demonstrate that it serves the purpose of achieving intersubunit crosslinking. But they fall short of addressing a key weakness of the approach: their data does not prove that the GluN1/2A and GluN1/2B crosslinks are identical, or even comparable. As stated by the authors, crosslinking is highly dependent on the distance, but it also depends on the physico-chemical microenvironment, which can greatly affect crosslinking efficiency. More specifically, the GluN1/2A and GluN1/2B interfaces around K178AzF will likely be similar, but not identical. Therefore, the authors cannot be certain that K178AzF crosslinks to

the same entity (side chain/backbone) on both the 2A and 2B subunits. Yet this would be an essential pre-requisite for them to make the comparisons that form the basis for their conclusions. Even minor differences in the angle of the crosslink or the location of the crosslinking site would potentially translate into major functional differences. But this would reflect differences in crosslink, rather than fundamental difference in subunit behavior.

This central issue extends to essentially all experiments, and is for example highlighted by the observed discrepancy in extent of photo-modulation in Fig3A/B. The authors explain this by the difference in basal P_o , but this is not the only possibility. The authors fail to prove that this is not caused by different type of crosslink, i.e. to a different side chain/backbone. Additionally, the kinetics look quite different and in neither case is the effect even remotely saturated over the time course of the experiment, which makes it very hard to compare, especially in light of the expected differences in crosslinking rate due to a non-identical microenvironment.

The authors need to provide mass-spectrometric evidence that the crosslinks occur between equivalent side chains to address the concern. Alternatively, the authors would need to significantly expand their cross-linking attempts at the interface by e.g. testing all 8 AzF-bearing variants in both GluN1/2A and GluN1/2B receptors. Only if the crosslinking patterns are the same in terms of extent and kinetics at all 8 sites, could the authors be reasonably confident that their system is behaving as they expect. In absence of such data this is not possible.

ii) The authors rely on MK-801 inhibition for testing P_o . This is a very useful screening method, but since the authors draw several conclusions about the P_o based on MK-801, it would have been valuable to see some single channel recordings. For example, the authors claim to conduct experiments in "high P_o mode" (line 249ff), but it is never experimentally shown that the P_o levels are comparable between GluN1/2A and GluN1/2B receptors following UV induced crosslinking, nor do the authors even provide any evidence for changes in P_o . Single channel recordings or noise analysis are required to back up these claims.

iii) Overall, the authors overstate how surprising it would be to find that the 2A and 2B subunits differ in their allosteric contribution (provided points i) and ii) above can be addressed). The 2A and 2B subunits are not nearly identical in sequence, so it is not surprising that they contribute differently. Instead, the authors should rather emphasize the strengths of their work, which is to show HOW they differ, as opposed to THAT they differ.

Minor comments:

The introduction could be more concise and there are some typos and awkward formulations, which makes an overall bad impression. As an example, line 49-50: "One released in the synaptic cleft, glutamate mediates fast neurotransmission by acting on ionotropic glutamate receptors (iGluRs) are subdivided into three main families". For example, it is unclear how the role a co-incidence detector is linked to control of "neuronal plasticity and higher brain function"; lines 77-89: very general statements, ultimately not clear how this is relevant to the present study.

A cartoon showing the receptor from top view of each extracellular layer, illustrating the positions for locking the dimers would help the reader.

The authors state that the GluN2A route relies on motions between two constitutive NTD dimers, which then translate into rearrangement within LBD D1-D1 interface. But what happens if the the GluN1/2A LBD inter-dimer is disulphide-locked? The authors only experimentally test LBD inter-dimer disulphide locking in GluN1/2B. The authors continue (line 465-469): "We note that domain swapping (also known as subunit cross-over), such that pairing of domains between the NTD and ABD layers involves different subunits, permits and explains why inter dimer motions within one layer translate into intra-dimer motions in the neighboring layer and vice versa.". An experiment showing what happens if the GluN1/2A LBD inter-dimer is locked should be included.

Line 354-357: For the in silico analysis the authors begin with the sentence: "Experimentally, the

potentiation induced by GluN1-K178AzF NTD photo-crosslinking was found to be highly state dependent in GluN2B, but not GluN2A, receptors. In the former, the photo-potentiation is inversely correlated with the receptor initial P_o , while in the latter it is basically state-independent (Fig. 3)." I agree that the photo-potentiation to some degree relies on the receptor P_o , but I would be careful with the wording and not call the photo-potentiation state-dependent.

Line 359-363: The simulations suggest that photo-crosslinking with AzF in GluN2B should be state-dependant and not occur in the non-active/inhibited state. However, this is not supported experimentally. Please comment on these contradicting observations. One could speculate if a smaller photocrosslinker than AzF would capture this type of state-dependancy?

Fig. 4b: I assume "N.I." is non-injected oocytes? Please add in the figure text. I miss some comments on the background bands.

Fig. 5b: In the figure text is written "mutant GluN1-N521C-L777C/GluN2B-E517C-L781C ", but I believe this should have been the GluN2A mutant.

Fig. 5c: To me the current trace seems to illustrate GNE in absence of zinc (observed potentiation approx.. 1.2-fold)? However, in the figure trace zinc is indicated to be present, not absent. In that case I would have expected to see a 3-fold potentiation. Please comment

Line 49: Spelling error, sentence does not make sense in its current form

Line 52: there is no such thing as "glutamate depolarization"

Line 58: L missing in GluN2A-D

Line 187: "...lobes are close one another locks...", should be: "...lobes are close to one another locks..."

Line 315: reference is missing

Line 333-335: Change sentence "...GNE3419 binding displacing..."

Line 467: I would prefer to only use one type of abbreviation for the ligand binding domain. Change ABD to LBD.

Line 1072: Spelling error "wilt-type"

Line 1097: Spelling error "wilt-type"

NCOMMS-21-01951
Revised manuscript
Reply to reviewers

We thank the three reviewers for their insightful and constructive comments on our manuscript. According to these comments, we have made several modifications to our work. Importantly, we have included new experimental data, including new mutant receptors combining disulfide crosslinks with photocrosslinkers, as well as patch-clamp noise analysis and single-channel recordings. These new results provide additional evidence supporting our initial results and conclusions. Hence, the disulfide crosslink experiments confirm the fact that GluN2A and GluN2B receptor use distinct allosteric routes by showing that the LBD inter-dimer interface is a crucial point for allosteric transduction in GluN2B, but not GluN2A, receptors. Furthermore, independent single-channel recordings and noise analysis establish that photopotential of GluN2B receptors following NTD photocrosslinking is fully accounted by an increase in channel open probability (P_o) with no change in unitary conductance, and that the photocrosslinked receptors are locked in a high P_o state.

We believe that altogether these revisions lead to an improved manuscript.

Reviewer #1 (Remarks to the Author):

The NMDA receptor is a subtype of ionotropic glutamate receptors that plays a key role in synaptic plasticity. Understanding NMDA receptor function and allosteric modulation is of great importance. Tian et al. use two-electrode voltage clamp in combination with unnatural amino acid-based photo-crosslinking and cysteine crosslinking to discover that GluN1/GluN2A and GluN1/GluN2B diheteromeric NMDA receptors take different conformational routes when gating is altered by negative and positive allosteric modulators. These findings show that stabilization in distinct conformations can achieve the same functional regulation. The manuscript is well written. The experiments are done carefully with appropriate controls and presented clearly. The findings are novel and advance the field substantially.

We warmly thank the reviewer for his positive appreciation of our work.

Two general points and several minor points should be addressed.

General Points

1) When comparing GluN2A and GluN2B receptor structures, the authors should take into consideration differences in stabilizing mutations, linker deletions and other sequence modifications that can affect the observed differences between GluN2A and GluN2B receptors structures. They should also comment on differences due to potential artifacts such

as receptor instability in cryo-EM studies (Mayer 2017; e.g. the discrepancy in results of Chou 2020 and Jalali-Yazdi 2018 for the splayed-open NTD configuration in presence of competitive antagonist or Jalali-Yazdi 2018 vs Zhang 2018 for pH effect on NTD.)

The reviewer is right that structural comparisons can be biased by the constraints of structure solving using X-ray or single-particle CryoEM approaches, by the type of constructs used, by the use of detergent instead of lipids to solubilize receptors, by biases in particle selection related to internal symmetry constraints or by over-interpretation of low resolution maps. Although we cannot guarantee that our analysis is free of biases, we did our best, as detailed below, to minimize potential interference. The main points are:

- We performed our analysis with a total of 34 structures of quasi full-length agonist-bound tetrameric GluN1/GluN2 receptors. Although these structures were solved independently by three different groups using distinct subunit constructs, little structural differences can be observed between structures obtained in similar conditions, either with X-ray crystallography (4PE5, 4TLL, 4TLM) or single-particle cryo-EM (5iov, 5fxh, 5fxi, 6whr, 6whs). This suggests limited technical biases.

- As mentioned by the reviewer, there are important discrepancies between NMDAR structures in complex with competitive antagonists obtained by different groups. This has led to a vivid debate about the relevance of certain of these structures which may not represent native conformations (due to technical issues regarding global stability of the multi-domain receptor complex). In our work, none of the selected NMDAR structures are in complex with competitive antagonists. Rather we analyzed only structures in the presence of agonists to focus on the transition between the active and allosterically inhibited states (all ligand-bound) in GluN2A and GluN2B receptors. We thus avoid the potential pitfalls linked to antagonist-bound preparations and structures.

- As also pointed by the reviewer, there are some discrepancies between the studies of Jalali-Yazdi et al. 2018 and Zhang et al. 2018 regarding the pH effect on GluN2A receptors conformation. However, the direct comparison of pH effects between the two studies is not so straightforward since the receptors were not prepared in the same conditions (buffer, detergent). Furthermore, the tested pH values differ (pH 8.0 and 7.4 with EDTA for Jalali-Yazdi 2018; pH 6.3 and 7.8 with EDTA for Zhang 2018). Accordingly, in our analysis, and after detailed inspection, we classified the structures obtained by Zhang et al. 2018 at pH 6.3 (6IRF) and pH 7.8 (6IRA) as close respectively to the Inhibited and 2KA states defined by Jalali-Yazdi et al. 2018. (Please note that we also now mention in the Methods section and Supplementary Table 1 that all GluN2A structural states were classified and named according to the nomenclature defined in Jalali-Yazdi et al. 2018).

In the revised manuscript, we have taken these points into account by modifying the Text at several locations (Results page 12, Discussion page 16 and Methods pages 23-24) and by adding a new Table (Supplementary Table 1). With these modifications, we i) better indicate how we selected the structures, ii) describe which modifications were introduced in the chosen structures and how it could affect our comparative GluN2A/GluN2B analysis, and iii) comment why the discrepancies that have been observed in the literature between different NMDAR structures obtained by different groups have little impact on our structural analysis. We have also introduced a new reference, Mayer 2017 Biophysical Journal, which nicely overviews potential methodological issues regarding structural analysis of iGluRs.

2) The authors should compare their iMODfit simulations with published MD simulations, e.g. opening and closing of the channel from Cerny (2019, 10.3390/biom9100546)

Thank you for pointing to the modeling work by Cerny and collaborators. In our revised manuscript, we now comment (and cite) this paper as well as the works by Zheng et al. (Biophysical Journal 2017) and Dutta et al. (Structure, 2015) which probe structural dynamics of NMDARs using coarse-grained elastic network analysis. Interestingly, both Zheng et al. and Cerny et al. report a similar large amplitude pore/TMD rotation event similar to what we observed in our modeling. What exactly this rotation represents in term of gating transition and activity state of the receptor remains still unclear (and it is not a central aspect of our work). We cannot extend the comparison much beyond this point however, in particular with the MD simulation of Cerny et al., since our groups have been investigating different transitions in the gating cycle of the receptor: Cerny et al. focus on the transition from the active to the resting state in GluN2B receptors, while we describe the transition between the active to the inhibited state in GluN2A receptors. At present, there is little evidence that these transitions share the same conformational pathways (see discussion in Mayer, 2017). We have modified our Text to include these elements and put in context our modeling studies in relation with the above-mentioned MD and NMA simulations. See revised Discussion page 16, end of 1st paragraph as well as page 18, also end of 1st paragraph.

Minor points

1) The authors should include all of the PDB identification codes in the text (and/or in a supplemental table) to make it clear to which structures they refer.

This is now done. As suggested, we have opted for a full Table (new Supplementary Table 1) that lists all the PDB identification codes mentioned in the manuscript. To these identification codes, we have associated several important piece of information (such as receptor's state, experimental conditions, resolution, etc...) to provide to the reader a clear overview of all the structures analyzed and discussed in our work. Please note that while constructing this Table, we spotted an error in the initial version of our manuscript when providing the total number of structures used for our structural analysis (which is 34 and not 29 as mentioned initially). This piece of information has been corrected in the revised manuscript.

2) I am aware that the NMDA constructs used in x-ray crystallography and cryo-EM are often called "full-length", however I would rather suggest to use e.g. "near full-length" throughout the manuscript since these receptors lack the whole C-terminal domain.

When mentioning 'full-length' receptors for the first time (page 3), we now precise that it is in fact 'near full-length' because of the CTD deletion. However, we also propose to keep the name 'full-length' as a shortcut in the rest of the paper (for reasons of simplicity). This is now explicitly stated in the Introduction (2nd paragraph of page 3).

3) Figure Legends: Missing Hill slope error value. Is there any reason for this?

No there was no specific reason for not including them. Values have now been introduced in the revised version.

4) Line 136: ">17 Å", is this Ca distance?

Yes. It is now clearly mentioned.

5) *Something funny happened to several parts of this sentence:*

Line 49-52: "One released in the synaptic cleft, glutamate mediates fast neurotransmission by acting on ionotropic glutamate receptors (iGluRs) are subdivided into three main families. AMPA, kainate and NMDA receptors. While AMPA and kainate receptors are primarily involved into glutamate depolarization of the postsynaptic membrane"

Typos corrected, thank you.

6) Line 58: "(GluN2A-D)

7) Line 299: "receptor"

8) Line 257: 7.0 ± 0.02 ; Line 261: 7.6 ± 0.01 and similarly throughout the text.

All typos corrected and missing numbers added throughout the Ms.

Reviewer #2 (Remarks to the Author):

In this interesting paper Paoletti and co-workers assess allosteric signalling in GluN2A vs GluN2B NMDAR heteromers. Using cross-linking approaches at strategic sites combined with patch clamp recordings they convincingly show differences in allosteric transmission from the distal NTD to the ion channel. Despite their overall structural similarity GluN2A and GluN2B NTD heterodimers show distinct dynamics which are transmitted differently through the LBD layer down to the ion channel. They also explore the rich pharmacology of these iGluRs to further our understanding of allosteric regulation of NMDAR by zinc and protons and differential regulation of the 2 NMDAR subtypes by these agents. This is a nicely conducted study that will be of interest to the wider community of neuroscientists, biochemists and structural biologists. The paper is clearly written and the figures document the story well.

We also warmly thank this reviewer for his positive evaluation of our work.

I have some additional comments.

1. The inter-LBD rolling, which seems a crucial point for GluN2B receptors (Fig. 5e-f) has not been tested for GluN2A heteromers. The latter are sensitive to intra-LBD dimer x-links, which seem to be ineffective in GluN2B receptors. Has an inter-LBD x-link been attempted in NR2As? This allosteric point comes up again in the modelling section at the end of the paper (Fig. 6), and would be good to include.

This is an excellent point and something we had not tested. We have now performed new experiments to investigate the influence of the inter-dimer LBD interface on GluN2A receptor allosteric signaling. For that purpose, we repeated the GluN2B experiment on GluN2A, same conditions, combining the inter-dimer LBD disulfide cross-link with the photoreactive GluN1-K178AzF mutant to probe for NTD allostery. For completeness, we also assessed allosteric signaling in GluN2A receptors combining the same AzF NTD mutant together with the intra-dimer disulfide crosslinks (another experiment that we had previously performed on GluN2B but not GluN2A receptors). The results, illustrated below (panel f, bar graph), are the following: constraining the inter-dimer LBD interface has no significant influence on NTD-mediated allosteric signaling (photo-potential of 1.29 ± 0.06 [n=6] for N1-C/N2A-C vs 1.38 ± 0.15 [n=8] for GluN1-K178AzF/GluN2A wt, n.s. $P=0.21$), while locking the intra-dimer LBD interface significantly reduces NTD-mediated photo-regulation (photo-potential of 1.18 ± 0.06 [n=12] for N1-CC/N2A-CC vs 1.38 ± 0.15 [n=8] for GluN1-K178AzF/GluN2A wt, ** $P=0.003$) (please also note that the mean value of 1.18 obtained for N1-C/2A-C receptors is not significantly different ($P=0.056$) from the value of 1.09 ± 0.10 (n=10) obtained on 'pure' wild-type GluN1/GluN2A receptors; see Fig 3b). This is in striking contrast with results previously obtained on GluN2B receptors, which showed that the inter-dimer but not intra-dimer LBD interface is critically involved (Figure 5). Thus, these new results provide further evidence and confirm our initial observations and conclusions that GluN2A and GluN2B receptors use distinct allosteric routes. They have been included in the revised manuscript (revised Fig. 5, panel f; and accompanying modifications in the main Text page 12, end of 1st paragraph).

2. could the authors comment on the impact of the GluN1 (exon5) splice variant in the allosteric routes they describe, ie does it behave the same or differently?

This is an interesting point. It is indeed well known that the presence of the GluN1 exon 5 (variant GluN1-1b) strongly decreased the pH sensitivity of GluN2B receptors, much less so of GluN2A receptors (from pH IC_{50} of 7.5 to 6.9 in GluN2B, vs 7.0 to 6.9 in GluN2A receptors; as published, among others, by the lab of Steve Traynelis; also unpublished data from our lab). That is to say, proton-wise, exon 5 appears to display GluN2B 'preference'. On the other hand, when assessing inhibition by extracellular zinc, exon 5 appears to affect GluN2A receptors (i.e. decrease zinc sensitivity) to a greater extent than GluN2B receptors (see Paoletti et al., J Neurosci 1997; Traynelis et al., J Neurosci 1998). So zinc-wise, exon 5 displays GluN2A 'preference'. From our current results, the proton allostery of GluN2B, but not GluN2A, receptors is mainly connected with NTD heterodimer packing. On the other hand, the zinc sensitivity of GluN2A, but not GluN2B, is strongly connected to the LBD intra-dimer interface. This is interesting because close inspection of the GluN1-1b/GluN2B receptor structure (Regan et al., Neuron, 2018), the sole NMDAR structure currently available with exon 5 (partially) resolved, shows that the exon 5 loop makes multiple inter-domain and inter-layer contact, compatible with both the GluN2A allosteric route (interaction with the LBD D1-D1 intra-dimer interface) and the GluN2B route (interaction with the neighboring NTD dimer). In other words, we currently think that exon-5 has the capacity to interfere with both allosteric routes, thus impacting the functionalities of both receptor subtypes. We plan to launch a series of experiments to test this idea. We would prefer not to mention exon 5 in the current work since we believe it is not critical for the scientific message

at this stage, and it would result in an inflated and over-densified manuscript (already pretty long!).

3. Supplementary Fig 2b also shows E185AzF, which is not mentioned. This mutant is shown in Fig 1c and not being potentiated by UV light and therefore it may make sense to remove it from Supplementary Fig 2b?

We think important to keep the information on the MK-801 inhibition kinetics of the GluN1-E185AzF/GluN2B mutant receptor not being affected by UV treatment because it fits well with the observation that UV has no effect on current amplitude on this particular mutant (in contrast to other GluN1-AzF mutants). However, the reviewer is right that the MK-801 results on the GluN1-E185AzF mutant were not mentioned in the initial version, while being presented in a supplementary figure, which was problematic. We now explicitly refer to this mutant when presenting the MK-801 results and cite the proper accompanying figures (see page 7, 1st paragraph).

4. line 193: NTD dynamics have also been described in FRET studies (Sirrieh R et al., JBC 2013) and in simulations (Dutta A. et al., Structure 2012), which should be mentioned here.

These two studies are now included in the manuscript (page 8, 1st paragraph).

5. Line 183: Supplementary Fig. 4 includes GluN2A and GluN2B but this section is only about GluN2B ?

We now refer to the GluN2A part of this Supplementary figure when first presenting the photopotential on GluN2A receptors (both in *Xenopus* oocytes and HEK cells) and the fact that the extent of potentiation is of much lower amplitude than that observed on GluN2B receptors. See main Text page 7, 1st paragraph, and new Supplementary Fig. 4.

Reviewer #3 (Remarks to the Author):

The work performed by Tian et al. employs unnatural amino acid crosslinking, Cys disulphide engineering, receptor pharmacology and in silico approaches to study the allosteric pathways employed by GluN2A and GluN2B subunits. The experiments are relatively simple, well designed and carefully executed, although they are built largely on repeating previously published mutations and experimental approaches. Based on their findings, the authors conclude that NTD-driven transduction pathways differ between GluN1/2A and GluN1/2B receptors.

Insight on subunit-specific functional coupling between NTD and LBD is of potentially great interest to elucidate how modulators differentially affect GluN1/2A and GluN1/2B receptors.

However, the findings of the present work are suggestive, but not conclusive with regards to the key point the authors are trying to make.

Major concerns:

i) The authors outline the strengths of their crosslinking approach and convincingly demonstrate that it serves the purpose of achieving intersubunit crosslinking. But they fall short of addressing a key weakness of the approach: their data does not prove that the GluN1/2A and GluN1/2B crosslinks are identical, or even comparable. As stated by the authors, crosslinking is highly dependent on the distance, but it also depends on the physico-chemical microenvironment, which can greatly affect crosslinking efficiency. More specifically, the GluN1/2A and GluN1/2B interfaces around K178AzF will likely be similar, but not identical. Therefore, the authors cannot be certain that K178AzF crosslinks to the same entity (side chain/backbone) on both the 2A and 2B subunits. Yet this would be an essential pre-requisite for them to make the comparisons that form the basis for their conclusions. Even minor differences in the angle of the crosslink or the location of the crosslinking site would

potentially translate into major functional differences. But this would reflect differences in crosslink, rather than fundamental difference in subunit behavior.

This central issue extends to essentially all experiments, and is for example highlighted by the observed discrepancy in extent of photo-modulation in Fig3A/B. The authors explain this by the difference in basal Po, but this is not the only possibility. The authors fail to prove that this is not caused by different type of crosslink, i.e. to a different side chain/backbone.

Additionally, the kinetics look quite different and in neither case is the effect even remotely saturated over the time course of the experiment, which makes it very hard to compare, especially in light of the expected differences in crosslinking rate due to a non-identical microenvironment.

The authors need to provide mass-spectrometric evidence that the crosslinks occur between equivalent side chains to address the concern. Alternatively, the authors would need to significantly expand their cross-linking attempts at the interface by e.g. testing all 8 AzF-bearing variants in both GluN1/2A and GluN1/2B receptors. Only if the crosslinking patterns are the same in terms of extent and kinetics at all 8 sites, could the authors be reasonably confident that their system is behaving as they expect. In absence of such data this is not possible.

We are grateful to the reviewer for highlighting the advantages of photocrosslinking approaches, which have shown useful in exploring the structural mechanisms of a variety of membrane receptors and ion channels. The reviewer is also right that we do not have direct identification of the photocrosslinking partner of our AzF mutants. Yet, our aim was not to prove that the GluN1/GluN2A and GluN1/GluN2B receptor crosslinkings are identical, necessarily involving the exact same atomic entities. Rather, our goal was to live monitor (in real time) the functional impact on receptor activity of constraining the NTD dimers in a compact conformation. We demonstrate using western blotting that the crosslinked bands can be detected by both anti-GluN1 and anti-GluN2B antibodies, establishing that the UV illumination results in an inter-domain, and not in an intra-domain or intra-subunit, crosslink. Given what is known on the photochemistry and the strict distance constraints for photocrosslinking to take place (Reddington et al., *Angewandte Chemie International Edition* 2013; Coin et al., *Cell* 2013; both cited in the initial manuscript), inter-subunit crosslinking can only occur if the two subunits come in sufficient close proximity, which is exactly what we

aimed for. Moreover, given the location of the AzF mutants along an α -helix that lines the surface of the NTD lower lobe and that faces the neighboring NTD lower lobe, the attachment of the GluN1 and GluN2 lower lobes one to another following illumination is the most likely event and parsimonious explanation. Finally, the functional readouts during UV-induced crosslinking between GluN1-K178AzF/GluN2A and GluN1-K178AzF/GluN2B are consistent. In both cases, we observed UV induced potentiation, i.e. currents significantly increase and crosslinking promotes the receptors entering into a high activity mode.

We acknowledge that using mass spectrometry analysis would be the ultimate approach to pinpoint the precise crosslinking sites. Until now however, and despite tens of photocrosslinking publications on membrane receptors and channels using engineered UAAs, we are not aware of any single study that has achieved precise identification of the photocrosslinking partner with atomic resolution. What has been achieved using UAA photocrosslinkers are identification of partners in large interactome studies, but at the level of whole peptides and not at the single residue level (e.g. Hino et al., J Mol Biol, 2011; Taupitz et al., Chemistry 2017; Wang et al., Chemical Communications, 2021; Liu et al., Advanced Biology, 2021). To the best of our knowledge, precise identification (with atomic resolution) of the crosslinking partner using photocrosslinking UAAs has been achieved only twice: first in a small soluble monomeric protein (intra-molecular link; Hoppmann et al., Angewandte Communications, 2014) and, more recently, between a GPCR and a β -arrestin complex (note that the UAA was introduced in the soluble β -arrestin, not the GPCR; Clark et al., Nat Chem Biol, 2020). For full-length multimeric membrane-embedded receptors studied *in cellulo*, as done in the present work, the bottleneck for the identification of UAA photo-crosslinked peptides is linked to multiples issues, including amount and purity of the protein, yield of photocrosslinking, just to mention a few. Moreover, without *a priori* knowledge about the crosslinking site in the partner protein, specific algorithms for comprehensive searching would have to be developed. The first search engine for interactome determination using genetically encoded chemical crosslinkers, OpenUaa, has been reported recently (Liu et al., Advanced Biology, 2021), although it does not include photo-crosslinking UAAs such as AzF or BzF.

So, as suggested by the reviewer, we decided to take the alternative approach, i.e. to expand our photocrosslinking screen and systematically compare GluN2A and GluN2B AzF mutants. For that purpose, we performed novel TEVC oocyte experiments to test the light-sensitivity of three additional GluN2A mutants, GluN1-R174AzF/GluN2A, GluN1-T182AzF/GluN2A and GluN1-E185AzF/GluN2A, so that eventually all 8 AzF bearing variants, 4 on GluN2A and 4 on GluN2B receptors, can be compared (see figure below and new Supplementary Fig. 7a). The extents of photomodulation are the following: 1.04 ± 0.04 (n=3) for GluN1-R174AzF/GluN2A; 1.18 ± 0.13 (n=13) for GluN1-T182AzF/GluN2A; and 1.30 ± 0.17 (n=13) for GluN1-E185AzF/GluN2A. As a reminder, the value obtained for GluN1-K178AzF/GluN2A receptors was 1.45 ± 0.23 (n=32) (Fig. 3). This GluN2A pattern, whereby GluN1-K178AzF stands out and yields the highest photomodulation, resembles that observed on GluN2B receptors (highest modulation among the four GluN2B mutants for GluN1-K178AzF/GluN2B; Fig. 1). The position GluN1-K178 thus appears to be a 'hot-spot' for AzF photocrosslinking both in GluN2A and GluN2B receptors. As suggested, we have also analyzed the kinetics of photocrosslinking and, as shown on the graph below, the time-course of UV photo-potentiation measured at pH 7.3 with the key position GluN1-K178AzF are not significantly different between the two receptor subtypes (mono-exponential fit; τ_{UV} of

182 ± 95 [n=8] for GluN2A receptors vs 119 ± 65 [n=23] for GluN2B receptors, P=0.12 Student's t-test; see Figure below and new Supplementary Fig. 7c). Regarding the kinetics, we would like to make two comments: i) the reviewer is right that for some cells, the UV-induced photopotential showed no apparent saturation over the time course of the experiment, rendering quantification of the kinetics difficult if not impossible. Such cells were excluded from our analysis (as explicitly mentioned in the revised Methods section) because we believe that they do not purely reflect the behavior of NMDAR currents. Indeed, it is well known that in *Xenopus* oocyte, long periods of NMDAR activation - as it is the case in these experiments (minutes) - can result in Ba²⁺ (or Ca²⁺) entry and slow activation of endogenous inward conductances (including Ca²⁺-dependent chloride conductances) (Leonard & Kelso, Neuron, 1990), adding up to NMDAR currents; ii) for most cells however, currents were satisfactorily fitted with mono-exponential fits and saturation of the UV effect was reached or closed to being reached. This was verified on many cells by checking that upon a second UV exposure, minimal effect on current amplitude is produced, thus indicating that a plateau was reached. Please also note that saturation of UV-induced effect could be observed on the vast majority of our HEK cell recordings, both for GluN2A and GluN2B receptors, where activation of endogenous currents following NMDAR activation is much less of an issue; iii) we don't really understand the underlying mechanism(s) dictating the time course of UV-induced photo-potential. Obviously, there are way slower than the photochemistry by itself, likely reflecting 'biological' mechanisms linked to NTD mediated conformational changes specific to NMDARs effects receptor related. Similar (slow) kinetics of NMDAR photomodulation were observed in our previous papers (see Zhu et al., PNAS 2014; Tian and Ye, Scientific Reports, 2016).

Overall, these novel results (comparison of all 8 AzF mutants; kinetics), together with our initial results, strongly suggest that GluN1-K178AzF photocrosslinking in GluN2A and GluN2B receptors occur in a similar physicochemical microenvironment and result in similar 'trapping' of the individual GluN1-GluN2 NTD dimers in a compact configuration. For all these reasons, we believe that our direct comparison between GluN2A and GluN2B photocrosslinking experiments is sound and valid. In our revised manuscript, we have included the new results presented below (Supplementary Fig. 7) and modified the text accordingly (see Results section pages 8-9; Discussion page 15, 2nd paragraph; and Methods section page 20 about kinetics analysis). In particular, in the revised Discussion, we added the following sentence (page 15): 'Although we did not identify the precise crosslinking sites in GluN1-K178AzF/GluN2A and GluN1-K178AzF/GluN2B receptors, the similarity in the photopotential profile and kinetics of AzF mutants (see Supplementary Fig. 7) as well the strict distance constraints for photocrosslinking (Reddington et al., 2013; Coin et al., 2013) point to similar physico-chemical microenvironment for both receptor subtypes, resulting in the trapping of individual NTD dimers in a compact conformation.'

ii) The authors rely on MK-801 inhibition for testing P_o . This is a very useful screening method, but since the authors draw several conclusions about the P_o based on MK-801, it would have been valuable to see some single channel recordings. For example, the authors claim to conduct experiments in “high P_o mode” (line 249ff), but it is never experimentally shown that the P_o levels are comparable between GluN1/2A and GluN1/2B receptors following UV induced crosslinking, nor do the authors even provide any evidence for changes in P_o . Single channel recordings or noise analysis are required to back up these claims.

As requested, in our revision, we have performed patch-clamp single-channel recordings as well as noise analysis experiments. The new results are included in the revised manuscript, both in the Main Text and as a full new Supplementary Figure (Supplementary Fig. 4 and see below).

For these experiments, we mainly focused our efforts on GluN1-K178AzF/GluN2B receptors, which occupy a central role in the whole story with an extensive characterization of their

properties and light-sensitivity at the macroscopic level. Using outside-out patches on HEK cells with single-channel resolution, we directly assessed the effect of photocrosslinking on unitary conductance. Using whole-cell noise analysis (from HEK cell recordings), and combined with the measurements of unitary conductances, we then derived measurements of absolute channel open probability (P_o) values before and after UV illumination (more details regarding recording conditions and calculations are available in the revised Methods section). Key results are following:

- The unitary conductance is not affected by UV induced crosslinking. Values of unitary currents are (see Figure below): 3.46 ± 0.25 pA ($n=9$) before UV vs 3.55 ± 0.26 pA ($n=9$) after UV ($P=0.28$; paired t-test).
- Channel open probability (P_o) is markedly increased by UV illumination and UV photocrosslinking promotes entry into a high P_o mode. Values are (see Figure below): 0.29 ± 0.09 ($n=6$) before UV vs 0.73 ± 0.12 ($n=6$) after UV (** $P < 0.001$; paired t-test).

These results clearly demonstrate that i) as expected for a modification involving an extracellular region of the receptor distal from the TMD pore domain, photocrosslinking increases the receptor activity through a P_o effect, not a unitary conductance effect ; ii) after photocrosslinking, the receptors are 'locked' in a high P_o state. These results are in full support with our initial conclusions and provide further and independent evidence that locking the individual NTD dimers in a compact conformation switches the receptors in a high activity mode.

In our revision, we now include the whole Figure shown below, also illustrating novel recordings of GluN1-K178AzF/GluN2A receptors, as a new Supplementary Figure (Supplementary Fig. 4) and present the associated results in the Results section (page 7, 1st paragraph). Please note that this new supplementary Figure replaces previous Supplementary Fig. 4.

iii) Overall, the authors overstate how surprising it would be to find that the 2A and 2B subunits differ in their allosteric contribution (provided points i) and ii) above can be addressed). The 2A and 2B subunits are not nearly identical in sequence, so it is not surprising that they contribute differently. Instead, the authors should rather emphasize the strengths of their work, which is to show HOW they differ, as opposed to THAT they differ.

The reviewer is raising here an interesting point that made us realize that we did not sufficiently explain nor illustrate what we meant about GluN2A and GluN2B receptor proximity. We also acknowledge that some of our formulation on the surprising side of GluN2A and GluN2B using different long-range allosteric routes were a bit overstated and not really necessary. Accordingly, we have toned down certain statements on this point (see end of the Introduction, deletion of 'contrary to expectations'; and beginning of the Discussion, deletion of '... a previously unforeseen...'). That said, we do think that the discovery that GluN2A and GluN2B uses distinct long-distance allosteric routes is surprising (at least it surprised us!) given what is known about GluN2A and GluN2B receptor sequences and structures (see paragraph below). And we do believe that it is interesting to give the context for the general reader before engaging to the core substance of our article, which is really about fundamental aspects of NMDAR machinery and how GluN2A and GluN2B differ in their allosteric coupling between domains and layers.

GluN2A and GluN2B receptors are indeed not identical in sequence as a whole, but they are the closest paralogs among all other NMDAR subunits. Furthermore, based on available atomic structures, the important domain-domain interfaces on which we are focusing in our work appear very close in terms of overall structural arrangements but also simply in terms of amino acid sequences (see Table below). In fact, the residues belonging to these interfaces are much more conserved than the whole receptor sequences, or even than all sequences encompassing the corresponding domains (76% identity & 84% homology for the NTD intra-dimer upper lobe-upper lobe interface; 94% and 100% for the LBD intra-dimer D1-D1 interface; Table below, numbers highlighted in red). So, it is surprising that GluN2A and GluN2B receptors are following different allosteric routes with differential engagement of domain-domain interfaces given their high degree of conservation. To better emphasize these aspects, we have now included in our revision the Table and illustration below as a new Supplementary Figure (Supplementary Fig. 1), and modified the main text accordingly (Introduction page 4, last paragraph). Moreover, we eliminated overstatements as previously mentioned.

C

	GluN2A vs GluN2B % identity/homology*	GluN2A vs GluN2C % identity/homology*	GluN2A vs GluN3A % identity/homology*	GluN2A vs GluN1 % identity/homology*
Entire sequence	54 / 65	45 / 53	22 / 34	23 / 37
LBD only	82 / 89	75 / 81	37 / 50	36 / 49
LBD Intra-dimer interface only (67 AA) – panel A	94 / 100	93 / 97	51 / 57	43 / 57
NTD only	53 / 69	36 / 48	20 / 32	18 / 31
NTD Intra-dimer interface only (37 AA) – panel B	76 / 86	48 / 65	24 / 35	16 / 24

*Blosum62

Minor comments:

The introduction could be more concise and there are some typos and awkward formulations, which makes an overall bad impression. As an example, line 49-50: “One released in the synaptic cleft, glutamate mediates fast neurotransmission by acting on ionotropic glutamate receptors (iGluRs) are subdivided into three main families”. For example, it is unclear how the role a co-incidence detector is linked to control of “neuronal plasticity and higher brain function”; lines 77-89: very general statements, ultimately not clear how this is relevant to the present study.

We apologize for the typos and unclear formulations and did our best to correct them. We also shortened several parts in the Introduction, in particular the section between lines 77-89. So it is now more concise. We do however believe that it is important for the general reader to introduce the notion that GluN2A and GluN2B receptors are likely to carry out different functions in the CNS and that their dysfunction, either hypo- or hyper-function, is associated with CNS diseases. It emphasizes the importance of better understanding subunit-specific NMDAR mechanism and regulation, in the global context of signaling diversity in this important family of neurotransmitter receptors.

A cartoon showing the receptor from top view of each extracellular layer, illustrating the positions for locking the dimers would help the reader.

A cartoon showing the positions for disulfide locking of NTD and LBD dimers has now been added to the manuscript (below and new Supplementary Fig. 6). We systematically refer to this Figure in the main Text when introducing the engineered disulfide bonds.

The authors state that the GluN2A route relies on motions between two constitutive NTD dimers, which then translate into rearrangement within LBD D1-D1 interface. But what happens if the the GluN1/2A LBD inter-dimer is disulphide-locked? The authors only experimentally test LBD inter-dimer disulphide locking in GluN1/2B. The authors continue (line 465-469): “We note that domain swapping (also known as subunit cross-over), such that pairing of domains between the NTD and ABD layers involves different subunits, permits and explains why inter dimer motions within one layer translate into intra-dimer motions in the

neighboring layer and vice versa.” An experiment showing what happens if the GluN1/2A LBD inter-dimer is locked should be included.

This is a very relevant point that has also been raised by reviewer #2 (point 1). As detailed in our reply to reviewer 2, we have performed additional experiments to test the effect of LBD inter-dimer disulfide lock, but also LBD intra-dimer disulfide lock, on NTD mediated allostery in GluN2A receptors. The results are clear-cut and provide further evidence that GluN2A and GluN2B receptors use distinct long-distance allosteric routes and domain coupling. The new results have been added in the main Figure 5 (panel f) and are discussed in the main Text (page 12, end of 1st paragraph).

Line 354-357: For the in silico analysis the authors begin with the sentence: “Experimentally, the potentiation induced by GluN1-K178AzF NTD photo-crosslinking was found to be highly state dependent in GluN2B, but not GluN2A, receptors. In the former, the photo-potentiation is inversely correlated with the receptor initial P_o , while in the latter it is basically state-independent (Fig. 3).” I agree that the photo-potentiation to some degree relies on the receptor P_o , but I would be careful with the wording and not call the photo-potentiation state-dependent.

The reviewer is right; our statement about state-dependence was misleading. We have modified the text for ‘Experimentally, the potentiation induced by GluN1-K178AzF NTD photocrosslinking was found to be highly dependent on the level of receptor activity for GluN2B, but not GluN2A, receptors.’

Line 359-363: The simulations suggest that photo-crosslinking with AzF in GluN2B should be state-dependant and not occur in the non-active/inhibited state. However, this is not supported experimentally. Please comment on these contradicting observations. One could speculate if a smaller photocrosslinker than AzF would capture this type of state-dependancy?

We do think that in GluN2B receptors, the AzF photo-crosslinking requires the local NTD dimer to adopt a compact conformation, i.e. having the GluN1 and GluN2B lower lobes in sufficiently proximity for the nitrene radical (generated by UV illumination of the azido moiety) to react with a nearby atomic partner. So, in that sense, we expect photo-crosslinking to be

'state-dependent'. But this does not mean that photo-crosslinking cannot happen in the resting state, as we observed experimentally. Indeed, there are good reasons to think (see Gielen et al., Nature, 2009; Tajima et al., Nature 2016; Romero-Hernandez et al., Neuron 2016) that the NTD dimers in particular can sample multiple conformations in resting conditions (i.e. without ligand binding), spontaneously oscillating between active/compact and inactive/relaxed conformations. From that perspective, we don't think there is an internal contradiction between our observations (simulations vs experiments). As a matter of fact, as illustrated in Figure 2, we were able to abolish the effect of photocrosslinking in GluN1/GluN2B receptors when we experimentally trapped the NTD dimer in an inactive conformation using disulfide crosslinks (a modification which is expected to prevent reversion to an active conformation of the NTD dimer). Thus, conformational mobility within the NTD dimer of GluN2B receptors is a critical factor for photocrosslinking, as predicted by our model.

The question regarding size of the photocrosslinker is an interesting one. In our mind, AzF can be categorized as a 'small' photocrosslinker (compared to BzF for instance). We speculate that smaller photocrosslinkers, assuming they exist, might not provide sufficient spatial reach to allow inter-domain attachment, and thus might turn functionally 'silent'. On the other hand, too long (and flexible) a photocrosslinker may impose little constraint on the internal conformational dynamics of the domains, with functional effects hard to interpret (and predict).

Fig. 4b: I assume "N.I." is non-injected oocytes? Please add in the figure text. I miss some comments on the background bands.

Yes, N.I is for non-injected oocytes. It is now included in the legend.

We regularly obtain faint background bands with certain antibodies, and that is the very reason for why we always show non-injected oocytes lanes for comparison purposes. We believe that these background bands are due to recognition of endogenous *Xenopus* oocyte proteins by primary and/or secondary antibodies. We systematically observed them, at variable positions depending on the antibodies used (see, our previous publications Riou et al., PLoS One 2012; Zhu et al., Nat Struct & Mol Biol 2013; Zhu et al., PNAS 2014; Esmenjaud et al.; EMBOJ 2019), as other expert groups in the field (see, for instance, Furukawa et al., Nature 2005). That said, they are usually faint and clearly different from the bands of interest (resulting from injected recombinant NMDAR subunits), thus not precluding to deduce firm conclusions about subunit expression and disulfide bond formation. In our revision, we have modified the figure and accompanying legend by adding an asterisk (*) next to the gels to point to non-specific background bands (see revised Fig. 4b).

Fig. 5b: In the figure text is written "mutant GluN1-N521C-L777C/GluN2B-E517C-L781C "; but I believe this should have been the GluN2A mutant.

Corrected. Thank you for spotting this error.

Fig. 5c: To me the current trace seems to illustrate GNE in absence of zinc (observed potentiation approx. 1.2-fold)? However, in the figure trace zinc is indicated to be present, not absent. In that case I would have expected to see a 3-fold potentiation. Please comment

The trace presented in the figure illustrates the effect of GNE on wild-type GluN2A receptors in the presence of extracellular zinc (i.e. on zinc bound receptors; as indicated by the labels/lines above the current trace for agonist, zinc and GNE applications). In this particular example, the extent of potentiation is indeed of ~3-fold (3.01, compare the current amplitude after and before GNE application). As shown on the bar graph, the extent of GNE potentiation is much lower in the absence of zinc (mean 1.27 fold), in agreement with the proposed allosteric coupling in GluN2A receptors between the NTDs and the LBD intra-dimer interface (where GNE binds). We are sorry if the illustrated trace was not clear enough. To avoid any confusion, we have now colored the whole trace in green to clearly refer to and match the color of the right bar (+ zinc conditions). We have also modified the legend of Fig. 5c to make the point clearer.

Line 49: Spelling error, sentence does not make sense in its current form

Line 52: there is no such thing as "glutamate depolarization"

Line 58: L missing in GluN2A-D

Line 187: "...lobes are close one another locks...", should be: "...lobes are close to one another locks..."

Line 315: reference is missing

Line 333-335: Change sentence "...GNE3419 binding displacing..."

Line 467: I would prefer to only use one type of abbreviation for the ligand binding domain. Change ABD to LBD.

Line 1072: Spelling error "wilt-type"

Line 1097: Spelling error "wilt-type"

Thank you again for spotting these typos and errors. They have all been corrected in the revised manuscript.

Reviewers' Comments:

Reviewer #1:

Remarks to the Author:

The authors did a good job of addressing the concerns raised in the review. The paper is very good and, from my point of view, is ready for acceptance.

Reviewer #2:

Remarks to the Author:

The authors have answered all my queries and are congratulated on a very nice study.

Reviewer #3:

Remarks to the Author:

The authors have done an excellent job addressing my concerns. I have no further comments.